# ISINGFORMER: AUGMENTING PARALLEL TEMPERING WITH LEARNED PROPOSALS

## ABSTRACT

Markov Chain Monte Carlo (MCMC) underlies both statistical physics and combinatorial optimization, but mixes slowly near critical points and in rough landscapes. Parallel Tempering (PT) improves mixing by swapping replicas across temperatures, yet each replica still relies on slow local updates to change its configuration. We introduce IsingFormer, a Transformer trained on equilibrium samples that can generate entire spin configurations resembling those from the target distribution. These uncorrelated samples are used as proposals for global moves within a Metropolis step in PT, complementing the usual single-spin flips. On 2D Ising models (sampling), IsingFormer reproduces magnetization and free-energy curves and generalizes to unseen temperatures, including the critical region. Injecting even a single proposal sharply reduces equilibration time, replacing thousands of local updates. On 3D spin glasses (optimization), PT enhanced with IsingFormer finds substantially lower-energy states, demonstrating how global moves accelerate search in rugged landscapes. Finally, applied to integer factorization encoded as Ising problems, IsingFormer trained on a limited set of semiprimes transfers successfully to unseen semiprimes, boosting success rates beyond the training distribution. Since factorization is a canonical hard benchmark, this ability to generalize across instances highlights the potential of learning proposals that move beyond single problems to entire families of instances. The IsingFormer demonstrates that Monte Carlo methods can be systematically accelerated by neural proposals that capture global structure, yielding faster sampling and stronger performance in combinatorial optimization.

## 1 INTRODUCTION

Large generative models are powerful at proposing structured candidates, but are often weak verifiers. This trade-off has inspired the concept of *generator-verifier* collaborations, particularly in reasoning tasks. A generator proposes many candidates at scale, and a separate rule-based verifier checks or corrects them. For example, in theorem proving, the generator is often a Transformer and the verifier is a proof checker (Trinh et al., 2024). The verifier provides guarantees that the generator by itself cannot.

In this work, we show that solving sampling and combinatorial optimization problems admit a similar construction. Markov Chain Monte Carlo (MCMC) can be viewed as a rule-based, principled verifier, satisfying detailed balance and known stationary distributions. Parallel Tempering (PT), also known as replica-exchange Monte Carlo (Hukushima & Nemoto, 1996), strengthens MCMC with nonlocal swaps across a temperature ladder while retaining local moves within each replica. Despite the nonlocal nature of PT, solving optimization or sampling problems in rugged landscapes remains difficult due to the long mixing and equilibration times.

We propose to couple a Transformer *generator* with an MCMC *verifier*. The generator which we call the **IsingFormer** is trained on equilibrium configurations of a given system and produces full-system proposals conditioned on the inverse temperature, $\beta$. The verifier is PT's Metropolis accept/reject step for swaps and its local Gibbs updates. Inspired by methods like Boltzmann Generators (Noé et al., 2019), IsingFormer provides *independent* nonlocal proposals that capture global structure. Then, PT accepts or rejects them with the Metropolis criterion, then continues to perform local updates and replica swaps.

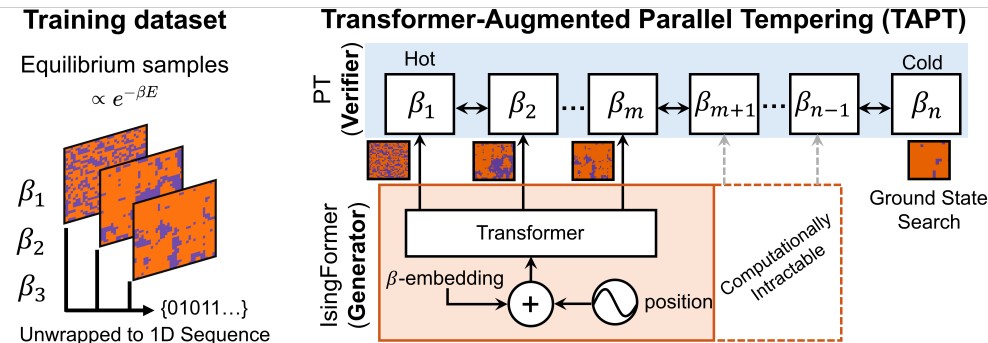

Figure 1: **The Generator-Verifier Framework for Parallel Tempering.** Equilibrium samples, generated via long-run Markov Chain Monte Carlo (MCMC), are used to train a decoder-only Transformer (IsingFormer generator). The model learns to produce full spin configurations conditioned on an inverse temperature, $\beta$. These configurations are then proposed as global moves within a Parallel Tempering (PT) framework, where they are accepted or rejected by a Metropolis criterion (PT verifier). This process augments PT's standard local updates and replica swaps, accelerating the search for low-energy states and escaping from local minima. Note that training data for high-$\beta$ (low-temperature) states is computationally intractable, so the coldest replicas are not augmented with learned proposals.

We call the resulting algorithm **Transformer-Augmented Parallel Tempering (TAPT)**. TAPT keeps PT's inverse temperature ($\beta$) and swap logic, but interleaves them with learned global moves (Fig. 1). Our hybrid algorihtm addresses two fundamental challenges. First, it addresses a neural network's lack of correctness guarantees by using PT as a principled verifier. Second, it overcomes the primary limitation of traditional MCMC that gets stuck in local minima by using the transformer to propose uncorrelated nonlocal moves. In our approach, we do not augment the coldest replicas with a transformer, since obtaining equilibrium training data at very high $\beta$ is intractable and amounts to solving the original problem.

We demonstrate that this generator-verifier pairing accelerates both sampling and optimization:

- **Sampling (2D Ising):** IsingFormer reproduces the free energy and magnetization curves of the 2D ferromagnetic Ising model and *generalizes* to unseen $\beta$ values, including the critical region. Even on conditioned configurations outside of the training set, injecting a single IsingFormer proposal replaces thousands of local MCMC updates by rapidly landing near equilibrium.

- **Optimization (3D Spin Glass):** TAPT lowers residual energy much faster than standard PT. A single warm start helps, but periodic proposals help even more.

- **Generalization across instances (Integer Factorization):** Encoding an invertible multiplier as a probabilistic Ising circuit and clamping the product yields families of semiprime instances. Trained on a subset of these, IsingFormer improves success rates on both seen and *unseen* semiprimes when plugged into TAPT.

Conceptually, TAPT belongs to the generator-verifier family of algorithms applied to sampling and combinatorial optimization. Neither standard PT nor a generator-only approach matches the tandem. This suggests a useful general template for sampling and combinatorial optimization.

## 2 RELATED WORK

Prior work on accelerating sampling or optimization problems with neural networks can be broadly categorized into two approaches: **(1)** training models to learn the full equilibrium (Boltzmann) distribution for sampling, and **(2)** training models to directly find low-energy states for optimization. Our work bridges these two approaches by using a model trained for equilibrium sampling as a proposal generator within a classic physics-based verifier framework.

## 2.1 LEARNED BOLTZMANN AND ISING SAMPLERS

There is a large body of work that learns equilibrium distributions to bypass long MCMC mixing times. Autoregressive Networks (VANs) fit the Boltzmann law and provide uncorrelated samples and free-energy estimates (Wu et al., 2019; Ma et al., 2024). RBMs and autoregressive models have been used as Metropolis proposal generators to accelerate MCMC (Huang & Wang, 2017; Wang, 2017; Wu et al., 2021; Nicoli et al., 2020). Diffusion generators have been benchmarked on 2D Ising models (Lee et al., 2025; Bae et al., 2025). Replica exchange between stacked RBMs has been shown to improve mixing time for MNIST, lattice proteins, and 2D Ising models (Fernandez-de Cossio-Diaz et al., 2024).

Beyond Ising models, Boltzmann Generators have been used to target molecular equilibrium and free energies (Noé et al., 2019; Invernizzi et al., 2022; Damewood et al., 2022). In lattice settings, normalizing flows serve as Metropolis proposal generators (Albergo et al., 2019; Kanwar et al., 2020; Boyda et al., 2021; Abbott et al., 2022).

**Difference to TAPT:** We adopt the "learned equilibrium sampler" as the generator but integrate it to a principled Monte Carlo framework (PT) with an explicit $\beta$ conditioning. The independent proposals are accepted with a Metropolis probability, supporting optimization and sampling by letting low-energy proposals propagate down the inverse temperature ($\beta$) ladder. Importantly, flows, diffusion models, RBMs, or alternative autoregressive samplers *complement* our approach: gains in generator fidelity will translate directly into higher acceptance and more effective nonlocal moves.

## 2.2 LEARNING FOR COMBINATORIAL OPTIMIZATION

Another body of work trains networks to minimize energy directly, rather than matching equilibrium distributions. Examples include variational neural annealing and annealed/tempered objectives for Ising and QUBO problems (Hibat-Allah et al., 2021; Ma et al., 2024; McNaughton et al., 2020; Zhang & Ventra, 2025). Transformers and other expressive architectures serve as variational ansätze in quantum Monte Carlo (Sprague & Czischek, 2024). Diffusion models target combinatorial families (MIS/MaxCut/QUBO), generalizing across instances within a class (Sanokowski et al., 2024; 2025). Real-valued "Ising-like" predictors learn a Hamiltonian whose minimum yields the task outputs on (spatio-)temporal graphs, with inference performed by energy minimization (Wu et al., 2024).

**Difference to TAPT:** These methods often directly optimize to find low-energy states and retrain models for different instances of a problem. TAPT instead learns an equilibrium-consistent generator and couples it with an explicit verifier. The optimization mechanism is entirely based on the powerful, physics-based Parallel Tempering framework. Moreover, we show that when the problem formulation allows conditioning to represent *different* instances, only one-time training is required and the generator generalizes.

## 3 VALIDATING THE GENERATOR: ISINGFORMER ON THE 2D ISING MODEL

Before coupling a generator to PT, we must show the generator can accurately learn equilibrium physics. We therefore evaluate IsingFormer on the 2D Ising model where thermodynamics are known exactly. Our evaluation has three parts that we will use later: **(1)** Unconditioned sample quality via free energy comparisons between IsingFormer and the exact solution, **(2)** *temperature generalization* across unseen $\beta$ and **(3)** conditional completion under clamped settings. These establish IsingFormer as a high-quality proposal generator for TAPT.

The 2D ferromagnetic Ising model is a stringent benchmark because its equilibrium statistics are known exactly via the Onsager solution at the thermodynamic limit (Onsager, 1944), and finite-size free energies can be computed exactly using the Kac-Ward determinant formalism (Kac & Ward, 1952) and related pfaffian methods. This enables a direct, quantitative comparison between model-generated samples and ground truth.

We trained a decoder-only transformer (IsingFormer) on equilibrium configurations of a $50 \times 50$ Ising grid with open boundary conditions, generated by long-run MCMC at several inverse temperatures $\beta$ (Appendix F). The objective is that the learned distribution $q_\theta(X)$ match the Boltzmann

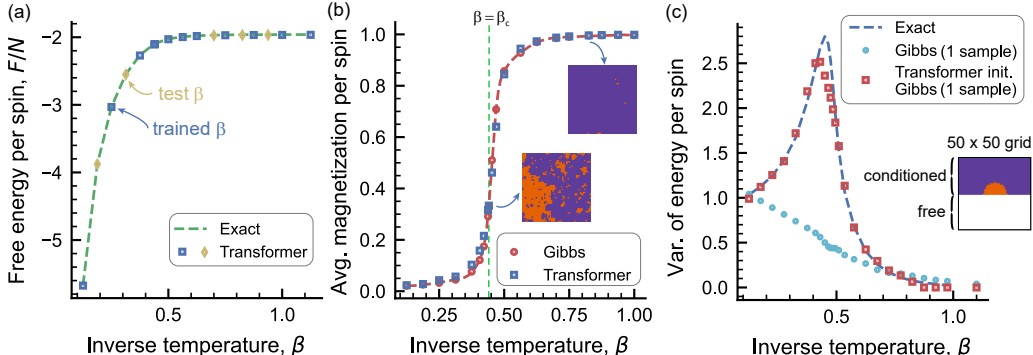

Figure 2: **Equilibrium learning on 2D Ising** ($50\times50$, open-boundary conditions): (a) Free energy per spin $f(\beta) = F(\beta)/N$ from Kac-Ward determinant formalism (exact, green dashed) vs. transformer-generated samples (yellow/blue). Diamonds mark trained vs. test $\beta$. The model reproduces $f(\beta)$ at trained points and interpolates to unseen $\beta$. (b) Magnetization vs. $\beta$, showing the transition near $\beta_c \approx 0.44$. Transformer samples (blue) match Gibbs sampling (red); insets show representative configurations obtained from the transformer. (c) Energy variance from the numerical derivative of the exact free energy (blue dashed) vs. single-sample estimates: Gibbs (cyan) and transformer-initialized Gibbs after *a single* update (red), markedly improving agreement. Insets show the half-clamped setup, unseen during training. The exact free energy (see Appendix A) was re-computed considering this clamped condition before computing the numerical derivative.

law $p_B(X)$, so that transformer samples reproduce the correct equilibrium observables. Critically, no free-energy terms are used at training time.

Fig. 2(a) shows that the transformer reproduces the exact free energy per spin, $f(\beta) = F(\beta)/N$ not only at the trained temperatures but also, critically, at unseen values of $\beta$. This demonstrates that the model generalizes across temperature and does not simply memorize the training set. The ability to interpolate in $\beta$ indicates that the model has learned nontrivial structure of the Boltzmann distribution rather than overfitting to individual training distributions. Interpolation is key for our TAPT algorithm, as it allows a single trained model to provide high-quality proposals for replicas across a range of temperatures while simplifying the training.

Beyond free energy, the transformer also captures ordering behavior. Fig. 2(b) shows that the average magnetization per spin obtained from transformer samples follows the expected sigmoidal curve, including the transition near the exact critical point $\beta_c \approx 0.44$. The ability to reproduce the sharp change in magnetization is especially significant because correlations become long-ranged at criticality, making this region difficult for purely local samplers. Inset configurations confirm that the generated states are physically consistent across ordered and disordered regimes, suggesting that the transformer is able to encode long-range correlations characteristic of the critical point.

The importance of transformer proposals becomes most evident in constrained settings. Fig. 2(c) considers the variance of the energy per spin, a quantity sensitive to sampling errors, under a half-clamped boundary condition. Pure Gibbs sampling from random initialization with a single update produces poor estimates, but transformer-initialized Gibbs sampling after just one sample, i.e. one update of all spins, aligns closely with the exact result obtained from the second derivative of the free energy. This result demonstrates that the transformer can act as a conditional sampler, filling in missing regions in a manner consistent with the underlying physics, something standard Gibbs sampling can only achieve after long equilibration. This result also illustrates the complementary roles of the verifier and generator: the transformer provides nonlocal moves that capture approximate global structure, while Gibbs sampling acts as a verifier to enforce detailed balance. In Appendix Fig. 5 shows this effect: while long Gibbs runs ($10^4$ updates) are required to reconstruct the correct boundary-induced structure, a single transformer sample followed by one Gibbs update already yields nearly indistinguishable configurations. In short, the trained transformer rapidly proposes near-equilibrium states, and MCMC corrects local defects and this is a useful feature for TAPT as we discuss next.

## 4  TAPT: TRANSFORMER-AUGMENTED PARALLEL TEMPERING

Building on the successful interplay between IsingFormer and Gibbs sampling, we propose to augment Parallel Tempering (PT) for optimization by incorporating equilibrium samples inferred by a transformer at different temperatures. Standard PT uses replicas of a system at increasing inverse temperatures $\beta_r$, with each replica sampling from its corresponding Boltzmann distribution at $\beta_r$ (Swendsen & Wang, 1986; Hukushima & Nemoto, 1996). To improve mixing, PT periodically attempts to swap the states of neighboring replicas with the acceptance probability:

$$P_{\text{swap}} = \min[1, \exp(\Delta\beta\Delta E)] \tag{1}$$

with $\Delta\beta = \beta_{r+1} - \beta_r$, and the energy difference between replicas $\Delta E = E_{r+1} - E_r$. Swaps are global moves that enable low-energy samples found at high temperatures to reach colder replicas.

Our main contribution is to introduce transformer-based global moves into PT, in which transformer samples are generated at the same PT temperatures to accelerate mixing and save thousands of local MCMC updates (Appendix Table 1). Thanks to the transformer's generalization capability across $\beta$ values, it can infer samples at new temperature points within the training range, providing flexibility to optimize the $\beta$-schedule. For a given replica $r$ at inverse temperature $\beta_r$, the corresponding transformer proposal is accepted with probability:

$$P_{\text{accept}} = \min[1, \exp(\beta_r \Delta E_r)] \tag{2}$$

where $\Delta E_r = E_r - E_r^T$ is the energy difference between the replica $r$ and the transformer proposal. Proposals with lower transformer energy are always accepted under the Metropolis criterion. This

---

**Algorithm 1** Transformer-Augmented Parallel Tempering (TAPT)

---

**Input:** Ising problem with energy $E$, number of replicas $N_R$, inverse temperatures $\beta_r$, number of swaps $N_{\text{swap}}$, samples per swap $M$, number of transformer inferences $N_T$.
**Output:** Low-energy configuration $(S^*, E^*)$
 1: Initialize replica states $S$
 2: Compute replica energies $E$
 3: $move \leftarrow 1$
 4: **for** $n = 1$ **to** $N_{\text{swap}}$ **do**
 5:     **if** $move = 1$ **then**
 6:         **for** $r = 1$ **to** $N_T$ **do**                                    ▷ Calling Transformer (**Generator**)
 7:             Provide context to Transformer                        ▷ From user or MCMC
 8:             Infer Transformer state $S_r^T$ at $\beta_r$
 9:             Compute Transformer energy $E_r^T$
10:             Compute $P_{\text{accept}} = \min[1, \exp(\beta_r(E_r - E_r^T))]$
11:             Assign $S_r \leftarrow S_r^T$ with probability $P_{\text{accept}}$
12:         **end for**
13:     **else if** $move = 2$ **then**                                    ▷ PT (**Verifier**)
14:         Swap even-odd replica pairs with probability $P_{\text{swap}}$
15:     **else**
16:         Swap odd-even replica pairs with probability $P_{\text{swap}}$
17:     **end if**
18:     **for** $r = 1$ **to** $N_R$ **do**                                    ▷ Sampling replicas
19:         Run MCMC for replica $r$ at $\beta_r$ ($M$ samples)
20:         Record last state $S_r$
21:         Compute energy $E_r$
22:     **end for**
23:     **if** $move \leq 2$ **then**                                    ▷ $move \in \{1, 2, 3\}$
24:         $move \leftarrow move + 1$            ▷ $move = 2, 3$: Replica swap (even-odd, odd-even pairs)
25:     **else**
26:         $move \leftarrow 1$                                    ▷ $move = 1$: Transformer–Replica move
27:     **end if**
28: **end for**
**return** Sample and energy of last replica $(S^*, E^*)$

---

rule is theoretically valid if the transformer samples from the Boltzmann distribution at $\beta_r$, an assumption supported by the transformer's high fidelity across the temperature range relevant for PT (see Fig. 6 in Appendix B).

The acceptance probability in Eq. 2 is the standard Metropolis criterion. While highly effective, it formally relies on the assumption that the proposal distribution is either symmetric or perfectly matches the target Boltzmann distribution. As a learned model, the *IsingFormer* is a powerful but imperfect approximator, so these conditions are not strictly guaranteed.

However, a unique strength of our autoregressive approach, over alternatives such as an encoder-decoder transformer, is that it provides a direct path to exact, unbiased sampling. Because the *Ising-Former* can compute the precise probability $P_{\text{model}}(m)$ for any given state $m$ due to its autoregressive nature, it enables the use of the full **Metropolis-Hastings (MH) correction**. This correction factor would precisely account for any biases in the generator, enforce detailed balance, and guarantee that the TAPT simulation converges to the true Boltzmann distribution. The focus of this work is on accelerating optimization and the uncorrected rule proves highly effective. As such, we do not implement this correction in our examples, however, this inherent strength motivates our choice for the decoder-only autoregressive transformer.

The Transformer-Augmented Parallel Tempering (TAPT) algorithm, outlined in Algorithm 1, alternates between global moves and local MCMC updates. The process begins by randomly initializing spin configurations for all replicas. For the first $N_T$ replicas, the transformer proposes new states (global move 1), which are accepted with probability $P_{\text{accept}}$. This transformer-based warm start is motivated by our previous experiment on the 2D ferromagnetic Ising model, which achieved orders-of-magnitude speedup compared to standard Gibbs sampling (Fig. 2c).

The coldest replicas are excluded from transformer proposals, as generating equilibrium samples at low temperatures is computationally intractable. Following the transformer proposals, each replica runs $M$ local MCMC updates. Then, with probability $P_{\text{swap}}$, even-odd replica pairs (e.g., (0,1), (2,3), (4,5)) attempt to swap their states (global move 2). After another $M$ local updates, odd-even replica pairs (e.g., (1,2), (3,4), (5,6)) swap (global move 3). This alternating sequence continues, with transformer proposals initiating each new cycle. After $N_{\text{swap}}$ global moves, the final state is read from the coldest replica.

## 5 TAPT FOR OPTIMIZATION

We now evaluate TAPT as an optimizer on two complementary tasks. First, we study a single 3D spin glass instance to isolate the speed and quality improvements from learned proposals. Second, we test integer factorization to assess generalization across problem instances sharing the same structure but different clamped outputs. The key question is whether *generator-based global proposals*, vetted by a principled Monte Carlo verifier, improve search in rugged landscapes. In both settings (spin glass and factorization), TAPT alternates learned global moves with local MCMC and replica swaps, exactly as in Alg. 1. As emphasized earlier, TAPT is *generator-agnostic*: any improved equilibrium generator (e.g., flows, RBMs, diffusion, or a stronger autoregressive model) can replace IsingFormer without changing the verifier or acceptance rules. Details of the training setup and wall-clock time are reported in Appendix F.

### 5.1 3D SPIN GLASS EXPERIMENT

We demonstrate Transformer-Augmented Parallel Tempering (TAPT) on the ground state search for a single instance of a 3D spin glass problem with $L^3 = 10^3$ spins shown in Fig. 3. We first optimize the temperature ladder using an adaptive scheduler (Isakov et al., 2015; Chowdhury et al., 2025; Mohseni et al., 2021; Nikhar et al., 2024), then run both PT and TAPT on the same schedule. Unless stated, we use 22 replicas, $M=5$ local updates per replica between global moves, and a total of $5\times10^3$ samples, where a sample corresponds to the update of all variables (sweep). In TAPT, the first 20 replicas are paired with the generator, the two coldest replicas are not augmented.

We report *residual energy* of the coldest replica, $\rho = \langle E - E_{\text{gnd}} \rangle / N$, averaged over 100 independent runs. The ground-state energy $E_{\text{gnd}}$ for this instance was estimated via extensive simulated annealing (best of 100 taking $10^6$ full sweeps each).

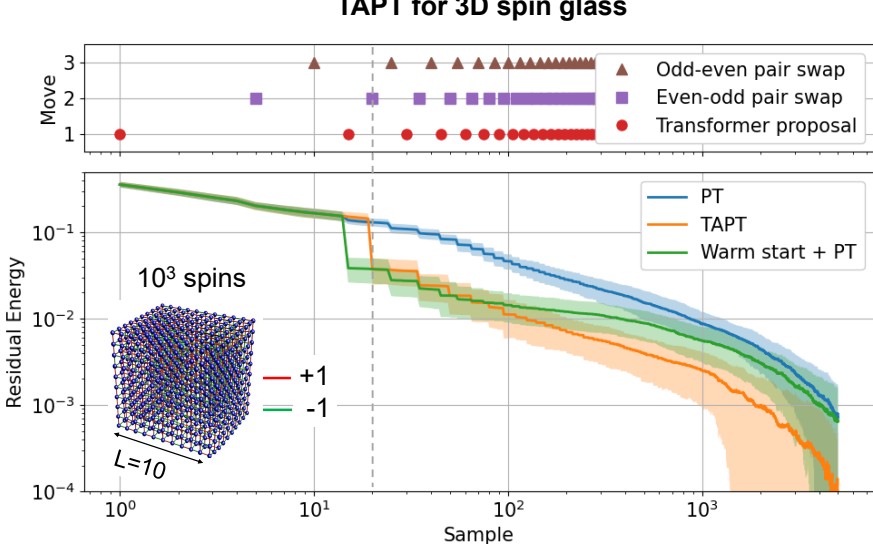

Figure 3: **TAPT on a 3D spin glass** ($L^3 = 10^3$ spins): We alternate local MCMC with three global moves: (*i*) generator proposals for the first 20 replicas, (*ii*) even–odd swaps, and (*iii*) odd–even swaps (Alg. 1). Global moves are spaced by 5 local updates. Both PT and TAPT use 22 replicas on the same adaptive $\beta$ ladder. Bottom: residual energy $\rho = \langle E - E_{\mathrm{gnd}} \rangle / N$ for the coldest replica, averaged over 100 runs (shaded: standard deviation). A one-shot generator warm start helps PT, however, periodic proposals (TAPT) yield larger gains. The dashed line marks the time when accepted generator proposals first reach the coldest replica via swaps.

Fig. 3 shows that TAPT produces a sharp early drop in $\rho$ when generator proposals begin to reach the coldest replica (dashed line), and continues to descend faster than PT as global moves and local updates alternate. A single generator warm start helps, but periodic proposals help substantially more (shallower slope in the log-log plot). In this instance, proposals are generated *independently* of the current replica state (no-context). Providing partial-state context to the generator did not improve performance in our experiments (Appendix C). As expected, acceptance diminishes at the cold end beyond the training $\beta$ range, but accepted moves at intermediate temperatures still propagate toward the coldest replica via swaps.

Note that the generator-verifier interplay is crucial: the generator supplies nonlocal proposals that escape basins, while local MCMC and replica exchange verify and refine these moves. Appendix D highlights the key role of MCMC refinement: removing local updates in a transformer-only variant collapses the accuracy. Here, we trained IsingFormer on a *single* spin-glass instance, and observed that the generator does not generalize to other instances (its proposals were entirely rejected), which is a common problem in neural optimizers. We acknowledge that transformer training time is not factored into optimization performance and in the absence of generalization, this is a serious limitation, as in the case of many neural optimizers. However, as we discuss next, TAPT demonstrates strong generalization to unseen problem instances (Fig. 4), when the original formulation allows expressing different problem instances via simple conditioning. In these cases, model training time will be amortized by inference, significantly improving mixing and saving thousands of Monte Carlo sweeps per problem.

## 5.2 GENERALIZING THE ISINGFORMER: CASE OF INTEGER FACTORIZATION

We next evaluate TAPT on a family of Ising instances that share structure but differ by instance. This is achieved by formulating the integer factorization problem as an invertible multiplier circuit, or as a circuit SAT instance (Borders et al., 2019). We next evaluate TAPT on a family of Ising instances that share structure but differ by instance: semiprime factorization via an invertible multiplier circuit. A logical multiplier $A \times B = C$ can be implemented with invertible AND gates and full adders (Andriyash et al., 2016; Aadit et al., 2022; Smithson et al., 2019; Pervaiz et al., 2019). Running

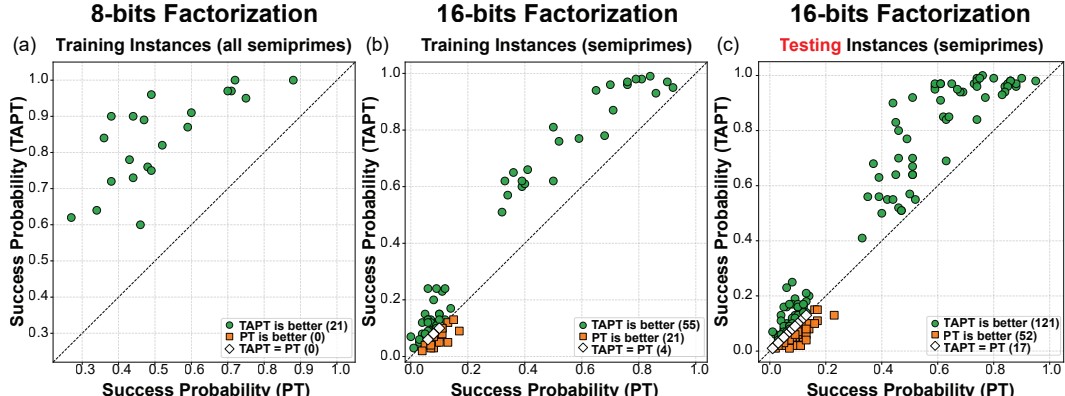

Figure 4: **TAPT on Semiprime Factorization.** We evaluate TAPT on families of factorization instances constructed from Ising multiplier circuits. In the 8-bit case, IsingFormer is trained on all 21 semiprimes and, when integrated into TAPT with 150 samples and $N_{\text{swap}} = 15$, outperforms PT on every training instance. For 16-bit factorization, IsingFormer is trained on a limited set of 80 semiprimes. TAPT with $10^4$ samples and $N_{\text{swap}} = 10^3$ achieves higher success probability than PT on about 70% of the training set. On 190 unseen 16-bit semiprimes, TAPT continues to outperform PT on roughly 64% of the test set. Together, these results show that TAPT accelerates optimization not only on training problems but also generalizes across new factorization instances.

the circuit in reverse by clamping the output $C$ yields a distinct Ising energy landscape for each semiprime $C$, with the factors $(A, B)$ corresponding to ground states. This setting naturally tests whether learned proposals *generalize across instances*. We choose integer factorization not as a high-performance algorithm (for which efficient number-field-sieve algorithms exist) but as a very hard optimization benchmark with an easily verifiable solution. Every semiprime output induces a new rugged optimization landscape while preserving the same underlying multiplier structure.

**8-bit semiprimes:** For 8-bit factorization ($N$=52 spins), there are 21 distinct semiprimes. We train IsingFormer jointly on all 21 instances at four $\beta$ values, then interpolate in $\beta$ at inference (Table 2 in Appendix F). We compare PT (8 replicas) against TAPT (8 replicas, generator proposals to the first 6 replicas), each run for 150 Monte Carlo sweeps with $N_{\text{swap}}$=15 swap attempts. The $\beta$-schedule is derived from the same adaptive procedure used in the 3D spin glass study, and we observe that the optimized schedule remains consistent across the family of 8-bits semiprime factorization problems. In TAPT, the output $C$ is provided to the IsingFormer as input tokens (along with carry in bits) that condition the generated proposals. Success is defined as satisfying $A \times B = C$ regardless of the internal multiplier configuration. As shown in Fig. 4(a), TAPT achieves higher success probability than PT on *every* training instance, indicating that learned nonlocal proposals consistently accelerate the search over this family.

**16-bit semiprimes (seen and unseen):** For 16-bit factorization ($N$=200 spins), we train on a subset of 80 semiprimes, again at four $\beta$ values, and reuse a single optimized $\beta$-schedule across instances. We used 15 replicas for PT and IsingFormer is providing proposals to the first 10 replicas. We compare PT (15 replicas) with TAPT (15 replicas, generator proposals to the first 10 replicas), each run for $10^4$ sweeps with $N_{\text{swap}}$=$10^3$. On the 80 training semiprimes, TAPT outperforms PT on $\sim 70\%$ of instances (Fig. 4b). Crucially, without any retraining, TAPT also improves over PT on $\sim 64\%$ of *190 held-out* semiprimes (Fig. 4c). This demonstrates generalization across instances that share the same circuit structure but differ in clamped outputs $C$. A detailed 16-bit factorization example is shown in Fig. 9 (Appendix E), where TAPT identifies the correct factors $A$ and $B$ at roughly twice the rate of PT.

Overall, these results demonstrate that learned global proposals can extend naturally across an entire family of problems, illustrating TAPT's ability to deliver acceleration beyond the training distribution.

## 6 CONCLUSION

We introduced Transformer-Augmented Parallel Tempering (**TAPT**), a hybrid algorithm that integrates a learned generative model (**IsingFormer**) into the principled framework of Parallel Tempering. By treating the transformer as a generator of global proposals and PT as a verifier, TAPT accelerates both sampling and combinatorial optimization. Our work bridges two distinct lines of research: those that learn equilibrium distributions for sampling and those that train neural networks for direct optimization. We demonstrate that a model trained only on equilibrium samples can dramatically improve a classic optimization heuristic.

Our main findings are threefold. First, on the 2D Ising model, IsingFormer successfully learns the equilibrium Boltzmann distribution, reproducing exact thermodynamic quantities like free energy and generalizing across temperatures, even in the challenging critical region. Second, when applied to a 3D spin glass, TAPT finds lower-energy states faster than standard PT, showing the power of learned global moves to escape local minima in rugged energy landscapes. Finally, and most significantly, TAPT demonstrates generalization across problem instances in the context of integer factorization. By training on a subset of semiprimes, IsingFormer learns proposals that improve success rates on both seen and unseen factorization problems.

This work leads to several key insights. The generator-verifier framework is a powerful template for leveraging the pattern-recognition strengths of deep learning without sacrificing the theoretical guarantees of traditional algorithms. Furthermore, TAPT is generator-agnostic, any advances in generative modeling for physical systems, be it with diffusion models, flows, or more powerful transformers, can be directly integrated into this framework to yield further improvements.

A promising direction for future work is to explore generalization over interaction terms ($J_{ij}$), which would enable a single model to tackle a broader class of Ising problems. The problem sizes and model scales used here are relatively modest, suggesting that significant performance gains may be unlocked by scaling up both the generator and the computational resources. Ultimately, TAPT shows that instead of replacing principled algorithms, generative models can serve as powerful co-processors, learning the global structure of a problem to guide a verifier that handles local refinement and guarantees correctness. Finally, we note that since a single learned proposal can replace thousands of local MCMC updates, the generator's interventions need not be frequent, even sparse proposals can dramatically accelerate convergence, effectively managing the computational cost of inference for problems of much larger size.

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

## A  CONTEXT EXPERIMENTS ON 2D ISING

In order to study constrained sampling and boundary-conditioned inference in the 2D Ising model, an exact baseline for the free energy under clamped boundary conditions is required. While the unconstrained Ising free energy has well-known closed-form solutions, the case with a full boundary row (or other partial boundary clamping) with a arbitrarily clamped condition is less standard. Deriving an exact expression for this conditional free energy serves two purposes: (1) It provides a ground-truth benchmark against which approximate MCMC or neural samplers can be evaluated and (2) It highlights the structural modification needed when external clamping is present - namely, the conversion of boundary interactions into effective fields, which can then be absorbed using the ghost-spin trick and expressed via the Kac-Ward/Kasteleyn–Pfaffian theorem.

### A.1  EXACT FREE ENERGY EXPRESSION FOR THE CLAMPED EXPERIMENT ON 2D ISING

In statistical physics, free energy $F$ is defined as

$$F(\beta) = -\frac{1}{\beta} \ln Z(\beta) \tag{3}$$

where $Z(\beta)$ is the partition function at a given inverse temperature, $\beta$:

$$Z(\beta) = \sum_{\{m\}} \exp\left(-\beta E(\{m\})\right) \tag{4}$$

with $E(\{m\})$ being the energy of the spin configuration $\{m\}$. For the 2D Ising case:

$$E(\{m\}) = - J \sum_{i=1, j=1}^{\substack{i=L_y-1 \\ j=L_x}} m_{i,j} m_{i+1,j} - J \sum_{i=1, j=1}^{\substack{i=L_y \\ j=L_x-1}} m_{i,j} m_{i,j+1} \tag{5}$$

Here we consider a $L_y \times L_x$ 2D Ising system with open boundary at all directions except the top boundary (corresponding to $i = L_y$) where the spins are clamped to some fixed configuration. Note that in the system under consideration, there are three categories of spin-spin connections, namely (1) clamped spin-clamped spin connections (corresponding to the horizontal connections on the top row; we denote the set of clamped spins as $F$), (2) clamped spin-free spin connections (corresponding to the vertical connections between the top row and the row just below the top row) and (3) free spin-free spin connection (all other horizontal and vertical connections; we denote the set of free spins as $U$). Let $\sigma_x \in \{\pm 1\}$ denote the fixed top spins with a prescribed clamping pattern

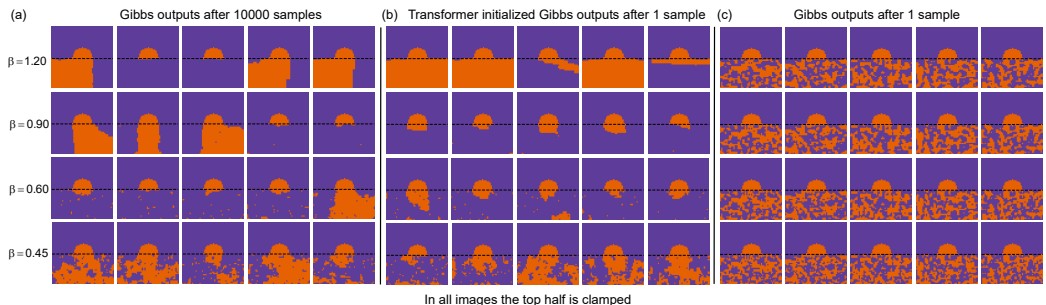

Figure 5: **Effect of transformer initialization under half-clamped boundary conditions:** Representative spin configurations on a $50 \times 50$ Ising grid with the top half clamped with a semicircular defect (orange: $-1$, purple: $+1$) at different inverse temperatures $\beta$. (a) Gibbs sampling after $10^4$ updates produces equilibrated configurations consistent with the boundary condition. (b) Transformer-initialized Gibbs sampling after only a single update rapidly produces configurations that closely resemble the long-run Gibbs outputs, demonstrating the benefit of transformer proposals. (c) By contrast, Gibbs sampling from random initialization after a single update fails to capture the correct structure. These results highlight how a single transformer sample can accelerate equilibration compared to purely local updates.

$\sigma = (\sigma_0, \ldots, \sigma_{L_x-1})$ (e.g. $+, -, -, +$). Then for the sake of simplicity, let us re-write the energy expression for a spin configuration $m = \{m_i\}_{i \in U} \in \{\pm 1\}^U$. as follows:

$$E(m; \sigma) = -J \sum_{\langle i,j \rangle \in E(U,U)} m_i m_j - J \sum_{\langle x,y \rangle \in E(F,F)} \sigma_x \sigma_y - J \sum_{\langle x,j \rangle \in E(F,U)} \sigma_x m_j, \tag{6}$$

where $E(A, B)$ collects edges with one endpoint in $A$ and the other in $B$. The conditional partition function that *sums only over the free spins* is

$$Z(\beta; \sigma) = \sum_{m \in \{\pm 1\}^U} \exp\big(-\beta\, E(m; \sigma)\big) \tag{7}$$

We are interested in the free energy of this system with the clamped row on top, i.e., $F(\beta; \sigma) = -\beta^{-1} \log Z(\beta; \sigma)$. Let us now define

$$Z_1 = \exp\big(\beta J \sum_{\langle x,y \rangle \in E(F,F)} \sigma_x \sigma_y\big) \tag{8}$$

This contains the contribution of the clamped row in $Z$ and is a constant with respect to free spin configurations, $m$. This yields

$$Z(\beta; \sigma) = Z_1 \prod_{e \in E(U,U)} \cosh\left(\beta J\right) \prod_{\substack{x \in F \\ \langle x,j \rangle \in E(F,U)}} \cosh\left(\beta J\right)$$

$$\sum_{m \in \{\pm 1\}^U} \prod_{\langle i,j \rangle \in E(U,U)} \Big(1 + m_i m_j \tanh\left(\beta J\right)\Big) \prod_{\langle x,j \rangle \in E(F,U)} \Big(1 + \sigma_x m_j \tanh\left(\beta J\right)\Big) \tag{9}$$

where we have made use of the identity

$$e^{K m_i m_j} = \cosh\left(K\right)(1 + m_i m_j \tanh\left(K\right)) \tag{10}$$

for bipolar variables $m_i$ and $m_j$. Next we apply an important trick, we re-write the partition function as

$$Z(\beta; \sigma) = Z_1 \prod_{e \in E(U,U)} \cosh\left(\beta J\right) \prod_{\substack{x \in F \\ \langle x,j \rangle \in E(F,U)}} \cosh\left(\beta J\right)$$

$$\frac{1}{2} \sum_{g \in \pm 1} \sum_{m \in \{\pm 1\}^U} \prod_{\langle i,j \rangle \in E(U,U)} \Big(1 + m_i m_j \tanh\left(\beta J\right)\Big) \prod_{\langle x,j \rangle \in E(F,U)} \Big(1 + g\, m_j \tanh\left(\beta J \sigma_x\right)\Big)$$

$$\tag{11}$$

where we have replaced the fixed-free factors with free-free factors by introducing a single 'ghost' spin $g \in \{+1, -1\}$. Note that applying this trick does not introduce any approximation and recovers the expression in Eq. (9) when summed over $g$. However, the double sum in Eq. (11) can be written as

$$\sum_{g = \pm 1} \sum_{m \in \{\pm 1\}^U} \prod_{\langle i,j \rangle \in E(U,U)} \Big(1 + m_i m_j \tanh\left(\beta J\right)\Big) \prod_{\langle x,j \rangle \in E(F,U)} \Big(1 + g\, m_j \tanh\left(\beta J \sigma_x\right)\Big)$$

$$= 2^{|U|+1} \,\Xi \tag{12}$$

and thanks to Kac-Ward/Kasteleyn-Pfaffian theorem on planar graphs (Cimasoni, 2010; Kasteleyn, 1961; Kardar, 2007) (note that the lattice even after introducing the ghost spin remains planar),

$$\Xi = \sqrt{\det\big(I - Q\big)}, \tag{13}$$

with $Q$ being the Kac-Ward transfer matrix indexed by directed edges $e = (u \to v)$ of the Ising lattice with the ghost spin $g$. Assuming two edges, $e = (u \to v)$ and $e' = (v \to w)$, the elements of the $Q$ matrix are written as

$$Q_{e,e'} = \begin{cases} C_K \, \exp\big(\frac{i}{2} \Delta\theta(e, e')\big), & \text{if } u \neq w, \\ 0, & \text{otherwise.} \end{cases} \tag{14}$$

Here $\Delta\theta(e, e')$ is the turning angle from edge $e$ to $e'$ under a fixed planar embedding, $C_K = \tanh(\beta J)$ for connections between free spins and $C_K = \tanh(\beta J \sigma_x)$ for free spin-ghost connections.

Combining the pieces in Eq. (9) we obtain the exact polynomial-time expression for free energy:

$$
\begin{aligned}
-\beta F &= \log Z(\beta; \sigma) \\
&= \sum_{\langle x,y \rangle \in E(F,F)} \beta J \, \sigma_x \sigma_y + \sum_{e \in E(U,U)} \log \cosh(\beta J) + \sum_{\langle x,j \rangle \in E(F,U)} \log \cosh(\beta J) \\
&\quad + |U| \log 2 + \frac{1}{2} \log \det(I - Q)
\end{aligned}
\tag{15}
$$

Finally, it can be noted that in the case of our experiment on 2D Ising with semicircular defect, all rows above the last clamped row of the top half also contribute to a constant term in the energy and hence can be included as a constant product term with the partition function (or as a sum in $\log Z$) expression obtained above.

**Numerical estimation of free energy:** Direct estimation of $Z$ from samples is hard because of the huge size of the state space. Hence, to estimate free energy at a given inverse temperature, we do the following:

$$
\frac{\partial(\beta F)}{\partial \beta} = -\frac{\partial \ln(Z)}{\partial \beta} = -\frac{1}{Z} \frac{\partial Z}{\partial \beta}
\tag{16}
$$

$$
-\frac{1}{Z} \frac{\partial Z}{\partial \beta} = \sum_{\{m\}} E(\{m\}) \frac{\exp(-\beta E(\{m\}))}{Z} = E_{\text{avg}}(\beta)
\tag{17}
$$

We first estimate $E_{\text{avg}}(\beta)$ from samples and then integrate over $\beta$:

$$
\beta F(\beta) = \text{const.} + \int_0^\beta E_{\text{avg}}(\omega) \, d\omega
\tag{18}
$$

The constant of the integration is found by evaluating $-\ln(Z(\beta))$ at $\beta = 0$. For a system of $N$ spins, there are $2^N$ terms to add in $Z$. At $\beta = 0$, each term contributes 1 to Z, yielding $Z(0) = 2^N$. So, the constant of integration is $-N \ln(2)$. If the system has some clamped spins, we replace $N$ with the number of free spins.

In this work, we do this numerical integration by linearly dividing the range from $10^{-3}$ to $\beta$ into 25 segments, taking 100 samples at each of the intermediate inverse temperatures and using trapezoidal rule for integration.

Table 1: Free energy per spin $F/L^2$ (top half clamped, open boundary at all sides, $L = 50$) at several inverse temperatures $\beta$. Estimates from transformers are averages of 100 samples. Gibbs estimates are computed from the averages of 100 independent runs.

| | | | **Free energy per spin**, $F/L^2$ | | | |
|---|---|---|---|---|---|---|
| $\beta$ | Exact | Transformer | Gibbs (10 samples) | Gibbs ($10^2$ samples) | Gibbs ($10^3$ samples) | Gibbs ($10^4$ samples) |
| 1.20 | $-1.92$ | $-1.92$ | $-1.83$ | $-1.91$ | $-1.91$ | $-1.91$ |
| 0.80 | $-1.92$ | $-1.93$ | $-1.85$ | $-1.91$ | $-1.92$ | $-1.92$ |
| 0.20 | $-2.74$ | $-2.77$ | $-2.74$ | $-2.74$ | $-2.74$ | $-2.74$ |

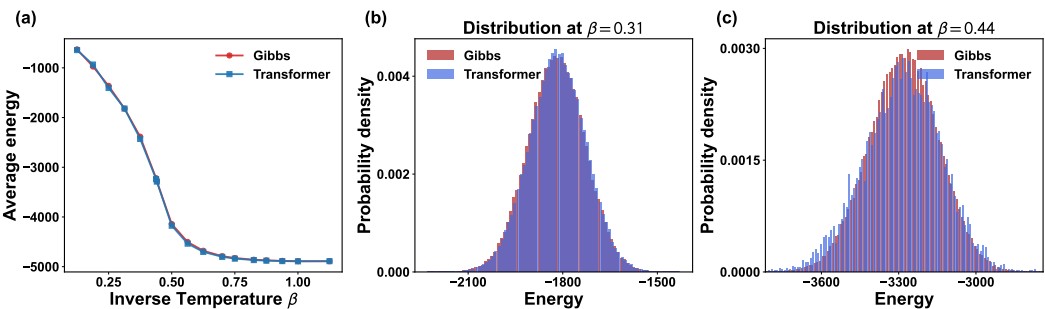

Figure 6: **IsingFormer Proposal.** (a) Average energy as a function of inverse temperature $\beta$, showing close agreement between Gibbs sampling and IsingFormer across a wide range of temperatures. (b) At $\beta = 0.313$ (below criticality), the transformer reproduces the equilibrium energy distribution with near-exact accuracy. (c) At the critical point $\beta \approx 0.44$, sampling becomes more challenging, and deviations emerge due to the intrinsic computational difficulty of criticality.

The second derivative of free energy also contains important information about the system and is related to the variance of energy:

$$
\begin{aligned}
\frac{\partial^2 (\beta F)}{\partial^2 \beta} &= + \left( \frac{1}{Z} \frac{\partial Z}{\partial \beta} \right)^2 - \frac{1}{Z} \frac{\partial^2 Z}{\partial^2 \beta} \\
&= + (E_{\text{avg}}(\beta))^2 - \sum_{\{m\}} E^2(\{m\}) \frac{\exp\left(-\beta E(\{m\})\right)}{Z} \\
&= -\text{var}(E(\beta))
\end{aligned}
\tag{19}
$$

In Table 1, we show the comparison of free energy estimates from Gibbs sampler and transformer with exact free energy value at several $\beta$. Appendix Fig. 5, shows sample configurations for the clamped 2D Ising example discussed in the main text.

## B    ISINGFORMER PROPOSALS

In Fig. 2, we demonstrated that the IsingFormer reproduces equilibrium statistics of the 2D ferro-Ising model across a wide range of inverse temperatures $\beta$. In particular, it captures not only thermodynamic limits but also system-specific quantities, such as the average energy of the $50 \times 50$ lattice as shown in Fig. 6(a). At low and moderate temperatures ($\beta < 0.44$), the transformer produces equilibrium-like samples with high fidelity, closely matching Gibbs sampling. Beyond the critical point, however, matching becomes significantly more challenging due to long-range correlation lengths, making it computationally intractable to generate equilibrium configurations at such $\beta$.

In TAPT, IsingFormer proposals provide accurate equilibrium-like samples across a broad range of $\beta$, effectively replacing thousands of local MCMC updates within each replica. At moderate temperatures, the IsingFormer closely matches Gibbs sampling and generalizes well even at interpolated $\beta$ values, enabling flexible scheduling of replicas. Yet, even if proposals are attempted in this cold replicas regime, they are naturally rejected by the Metropolis acceptance rule, which should not downgrade the TAPT acceleration at moderate $\beta$.

## C    CONTEXT AND ABLATION STUDY

An intriguing question in the TAPT algorithm is that of context. In the main sections, all the experiments were performed without providing any context to the transformer (line 7 in Alg. 1). As MCMC in each PT replica cultivates a Markov chain by performing local updates, one could ask whether transformer proposals could use this context, by being conditioned on parts of the chain

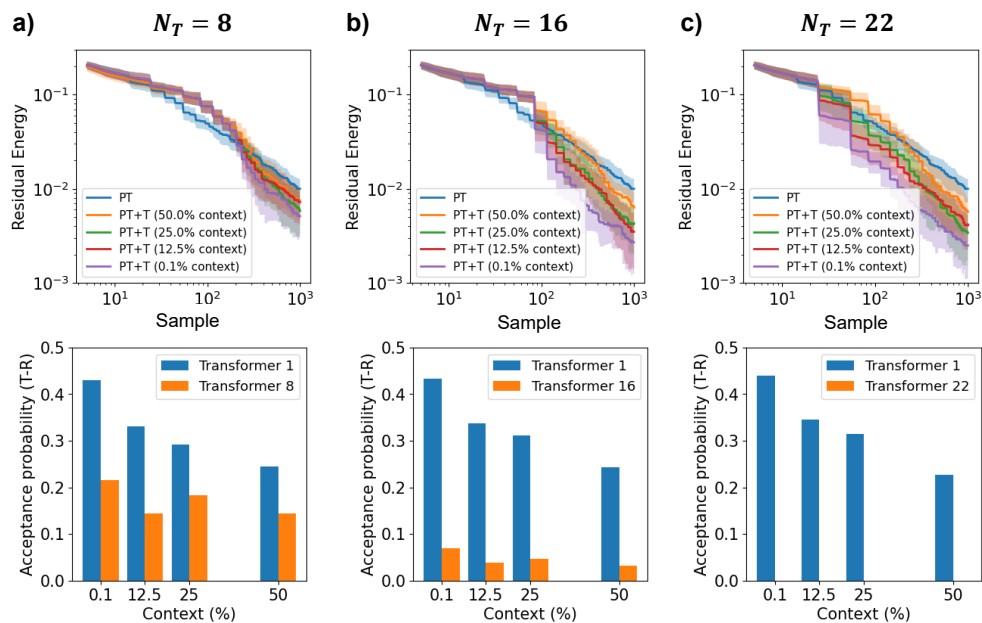

Figure 7: **Transformer context and ablation study** for a 3D spin glass problem with $10^3$ spins and 22 replicas, where we vary the number of transformer inferences at colder temperatures ($N_T$ is the number of inferred $\beta$ values). The top panel shows the residual energies of baseline parallel tempering (PT) and our hybrid approach with transformers (TAPT). The mean residual energies are estimated with 20 independent trials (shaded: standard deviation). The bottom panel shows the acceptance probability of the first and last transformer proposals.

currently running on a replica. To study this question, we vary the context provided by replicas to the transformer corresponding to a portion of the current replica state (starting from the first spin), fed to the transformer before each proposal. We observe that reducing the transformer context consistently improves the performance (Fig. 7). This might seem counter-intuitive, but it highlights a fundamental difference between using a generator for sampling versus optimization. For optimization, the goal is to escape local energy minima. Providing the current stuck state of an MCMC replica as context could bias the generator towards proposing new configurations within the same energy basin. The generator's primary strength in TAPT is its ability to produce uncorrelated global proposals, which are more likely to land in entirely different and potentially lower-energy regions of the state space. This suggests that for optimization tasks, a no-context approach may be superior as it maximizes the exploratory power of the learned proposals.

Next, we investigate the effect of removing transformer proposals at lower temperatures. Specifically, we vary $N_T$ from Algorithm 1, corresponding to the first $N_T$ $\beta$-values inferred by the transformer in TAPT. We consider the 3D spin glass experiment from Section 5.1 with 22 replicas and 20 $\beta$-points inferred by the transformer (the smallest $\beta$-values). Fig. 7 show the results with $N_T = 8$, 16, and 22. Using more transformer inferences reduces the residual energy, thus improving the performance. However, the improvement is marginal beyond the transformer training range for $\beta$ (maximum $\beta = 2$), for which transformer proposals are rarely accepted, as shown in the bottom plots.

## D  INTERPLAY BETWEEN MCMC AND TRANSFORMER IN TAPT

TAPT harnesses transformer proposals to accelerate PT with new global moves. Here, we demonstrate that TAPT's superior performance stems from a successful interplay between the transformer ("generator") and local MCMC steps that leverage the transformer samples ("verifier"). Only inferring the transformers and disabling local MCMC improvements collapses TAPT's success probabil-

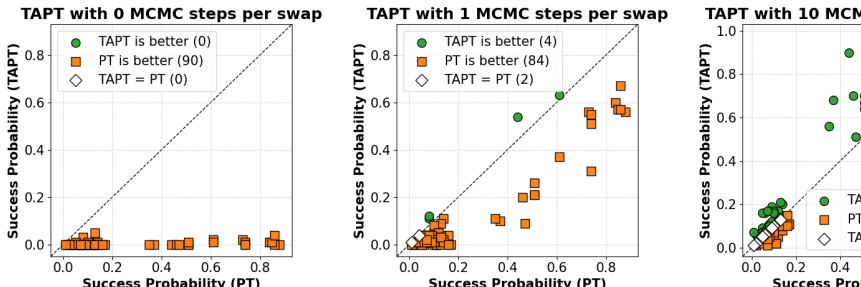

Figure 8: **Impact of local MCMC steps in TAPT**. We vary the number of MCMC samples $M$ between replica swaps and transformer proposals in TAPT (Algorithm 1) for 16-bit semiprime factorization. TAPT results are compared against baseline PT with fixed $M = 10$ and 90 test instances, 100 trials each. Left plot: removing MCMC refinement of transformer proposals collapses the success probability. Middle: restricting to a single MCMC step between global moves is worse than baseline PT for most test instances. Right: enabling $M = 10$ MCMC steps in TAPT significantly improves baseline PT for the majority of test instances. This highlights the efficient interplay between the transformer acting as a generator and classical MCMC, which verifies and builds upon transformer proposals.

ity. The results are shown in Fig. 8 for the 16-bit factorization problem of Section 5.2. In this study, we vary the number of local MCMC steps for all replicas. The x-axis corresponds to the success probability of the baseline PT with $10^4$ samples and $M = 10$ steps between each swap. Restricting the replicas to a single MCMC local step ($M = 1$) between global moves severely impacts the performance (middle panel), while allowing $M = 10$ steps outperforms PT for the same number of samples (right panel).

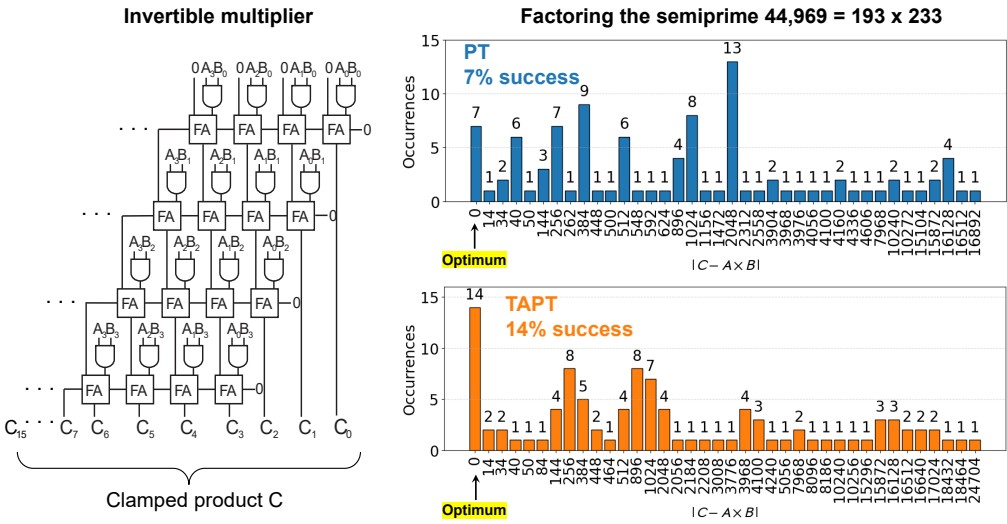

Figure 9: **Example of 16-bit factorization with PT and TAPT**. The circuit on the left is a 16-bit multiplier composed of invertible AND gates and Full Adder (FA) modules. Clamping the output $C$ and finding the ground state energy of the circuit solves the factorization problem. Both PT and TAPT are run 100 times to factor the semiprime $C = 44,969 = 193 \times 233$ for $10^4$ samples, with $M = 10$ samples between swaps and transformer proposals. Histograms show the distance of the measured product $A \times B$ from the clamped product $C$. PT and TAPT have 7% and 14% of success, respectively.

Table 2: Trained IsingFormers

| Instance Type | Size | Number of Samples/Instances | Number of Instances | $\beta$-range | Mixing Samples | Minutes/Epochs @ Batch | Max Epochs |
|---|---|---|---|---|---|---|---|
| Ferromagnetic | $50\times50$ (2500) | $14 \times 10^5$ | 1 | [0.125, 1.125] | $10^4$ | 40 @ 64 | 100 |
| 3D Spin Glass | $L{=}10$ (1000) | $31 \times 10^5$ | 1 | [0.125, 2.0] | $10^4$ | 15 @ 64 | 100 |
| Factorization | 8-bit (52) | $4 \times 10^5$ | 21 | [0.3, 1.0] | $10^4$ | 1 @ 128 | 150 |
| | 16-bit (200) | $4 \times 10^4$ | 80 | [0.3, 1.0] | $10^3$ | 4 @ 128 | 150 |

## E    FACTORIZATION CIRCUIT WITH INVERTIBLE LOGIC GATES

We detail the setup of the integer factorization experiment from Section 5.2. The transformer is trained on samples from the invertible multiplier circuit shown in Fig. 9. The circuit is built from invertible AND gates and Full Adder (FA) units, each implemented as coupled Ising spins, with coupling coefficients taken from (Aadit et al., 2022). In forward mode, the circuit performs multiplication $C = A \times B$ by clamping the corresponding input bits. In this work, we operate the circuit in backward mode to factor a given product $C$: when $C$ is clamped, the ground states of the circuit encode valid factors $A$ and $B$, which are recovered from the coldest replica.

Fig. 9 illustrates an example of 16-bit semiprime factorization with $C = 44,959 = 193 \times 233$, where PT and TAPT are each run for 100 trials ($10^4$ samples per trial), achieving success probabilities of 0.07 and 0.14, respectively.

## F    TRAINED ISINGFORMERS AND DATASET

The IsingFormer is trained by binary cross-entropy loss function on equilibrium samples obtained from long-run MCMC; empirical mixing times for all experiments are reported in Table 2. The model is a decoder-only Transformer with causal masked self-attention, sinusoidal positional encodings, and a learnable inverse-temperature embedding $e_\beta$ that conditions generation of spin tokens. We model spin strings autoregressively via next-token prediction and use a compact configuration with $d_{\text{model}} = 64$, $h = 2$, FFN $= 128$, and $L = 2$ layers, totaling in around $67 \times 10^3$ trainable parameters.

In the 2D Ising and 3D spin-glass instances, the training datasets consist of unconditioned equilibrium samples at different $\beta$ values. No clamping is applied, and the model learns to reproduce full-spin configurations drawn from the Boltzmann distribution.

For semiprime factorization circuits, the output product bits $C$ are clamped as fixed input tokens, together with the carry-in bit (always set to zero). The model is therefore trained conditionally, generating valid spin configurations consistent with the fixed $C$. Training is performed across multiple $\beta$ values for each clamped instance. For testing, as reported in Fig. 4(c), we clamp the IsingFormer to new $C$ values that were never provided during training.

Training is performed on an *NVIDIA RTX 6000 Ada* GPU. The IsingFormer is trained for a fixed maximum number of epochs, as reported in Table 2, and the final model is selected based on the checkpoint with the lowest validation loss. In terms of wall-clock time, the $50 \times 50$ 2D Ising model required roughly 3 days of training ($\sim$4000 minutes), while the $L = 10$ 3D spin glass completed in about 1 day ($\sim$1500 minutes). Factorization tasks were significantly faster: the 8-bit models trained in about 2.5 hours ($\sim$150 minutes), whereas the 16-bit models trained in about 10 hours ($\sim$600 minutes).

## G    LLM USAGE STATEMENT

LLMs were used to polish the presentation and writing of this contribution.

