# Supplementary Figures
# IsingFormer: Augmenting Parallel Tempering With Learned Proposals

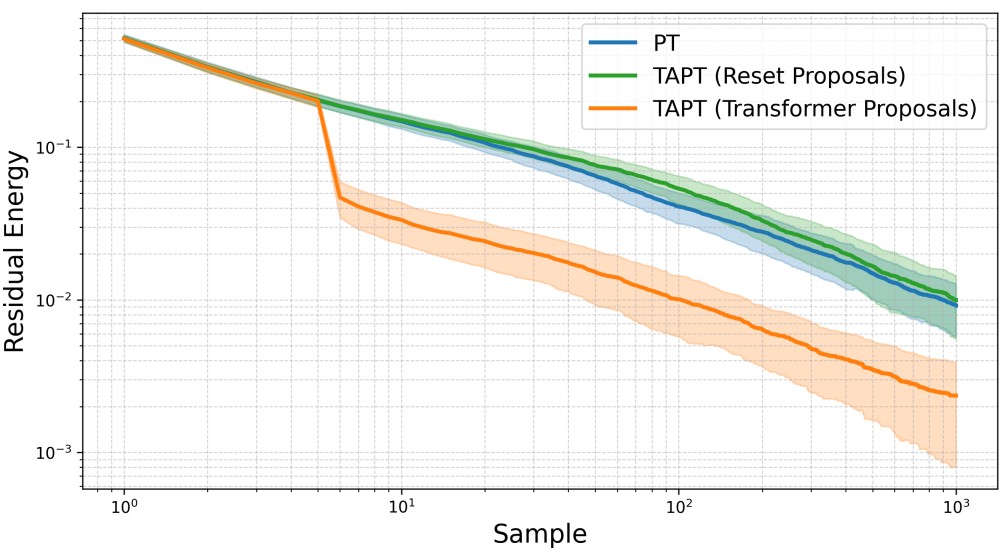

FIG. 1. TAPT with periodic global resets. We test TAPT scheme in two settings, one where we use transformer to give proposal to PT, for the other setting we replace the transformer proposal with random states (acting as a proposed reset). Because of the Metropolis acceptance rule, no reset is accepted, and proposing random state ends up to be similar to baseline parallel tempering.

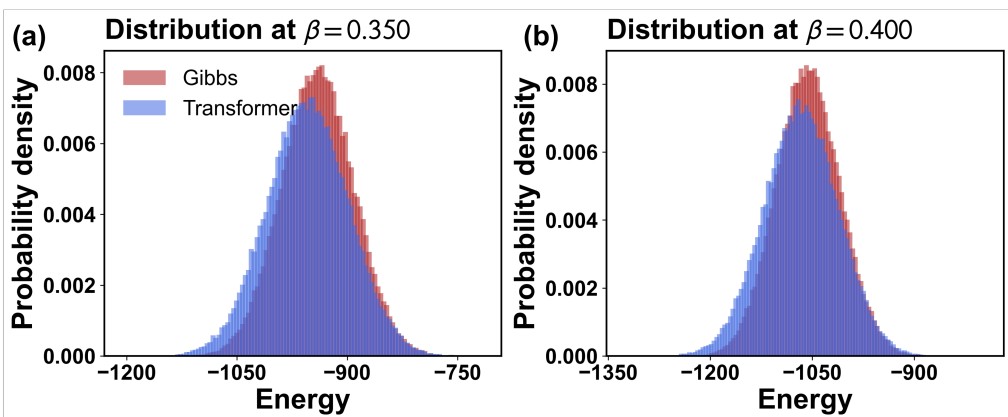

FIG. 2. **Generalization over $\beta$ on 3D spin glass**. Transformer generated samples at two $\beta$ values that were not seen during training. We observe reasonable overlap between the MCMC samples and the transformer samples.