# OpenReview forum: "IsingFormer: Augmenting Parallel Tempering with Learned Proposals"
_ICLR.cc/2026/Conference — Submitted to ICLR 2026_

### Official Review · Reviewer_bXSR · 2025-10-21

**Soundness:** 3
**Presentation:** 3
**Contribution:** 1
**Rating:** 4
**Confidence:** 4

**Summary:**

The authors propose IsingFormer, a Transformer trained on equilibrium configurations of Ising systems to generate entire spin configurations conditioned on temperature. Integrated into Parallel Tempering (PT) — a standard MCMC approach — these learned global proposals act as rapid, uncorrelated moves, which are then verified or rejected by the Metropolis criterion. This combination, termed Transformer-Augmented Parallel Tempering (TAPT), preserves PT’s theoretical guarantees while improving its efficiency by replacing thousands of local updates with a few learned global ones.
Empirical results show that IsingFormer reproduces key thermodynamic observables and generalizes to unseen temperatures in the 2D Ising model, significantly reducing equilibration time. When applied to harder optimization problems such as 3D spin glasses and integer factorization encoded as Ising systems, TAPT consistently finds lower-energy solutions and generalizes to unseen problem instances.

**Strengths:**

The paper is clearly and well written, with a well-organized structure that makes it easy to follow. The related work section is particularly useful, as it not only provides an overview of prior research but also clearly explains how the proposed approach differs from existing methods. The problem addressed — improving the mixing efficiency of standard MCMC methods and accelerating sampling and optimization — is highly relevant on its own. Moreover, the authors go a step further by applying their framework to combinatorial optimization problems via the Ising model, thereby broadening the potential impact and applicability of their work.

**Weaknesses:**

From my understanding of the paper, I have the feeling the novelty is more limited than what the authors claim.
Specifically, the authors propose a transformer-based autoregressive framework, conditioned on the inverse temperature $\beta$ capable of generating configurations at different temperatures. Furthermore, the combine this with Parallel Tempering to reach a far more efficient sampling routine.  Furthermore, limitations of the proposed method are currently not thoroughly discussed explicitly in the paper.

However, both these novelties seem to have been proposed in the past, see for example:
- [Temperature steerable flows and Boltzmann generators](https://journals.aps.org/prresearch/abstract/10.1103/PhysRevResearch.4.L042005)
- [Skipping the Replica Exchange Ladder with Normalizing Flows](https://arxiv.org/abs/2210.14104)
- [Conditional normalizing flow for Markov chain Monte Carlo sampling in the critical region of lattice field theory
](https://journals.aps.org/prd/abstract/10.1103/PhysRevD.107.014512)

**Questions:**

- Related to the weaknesses pointed above, I'd kindly ask the authors to elaborate more about their framework and its unique property which makes it different, and potentially more performant, compared to the approaches listed above.
- The conditioning of the transformer on the inverse temperature seems to suggest that it is only able to interpolate between different values of $\beta$. Nevertheless, [prior works](https://journals.aps.org/prd/abstract/10.1103/PhysRevD.107.014512) seems to report that extrapolation is also possible. I was therefore wondering if this is an inherent limitation of transformers or if the authors have not tried extrapolation at all.
- Why do the authors picked an autoregressive transformer as a generative model instead of a Diffusion model, autoregressive neural network (e.g., PixelCNN) or a normalising flow?
- This approach requires data for training while approaches like Boltzmann generators are data free. What is the advantage then?
- Section 4 mostly focus on the metropolis step within the TAPT algorithm which ensure to retain detailed balance and other properties typical of MCMC methods. The analysis therein seems very similar to the idea proposed by [Nicoli et al. (2020)](https://journals.aps.org/pre/abstract/10.1103/PhysRevE.101.023304). Am I seeing it right or does your method bring some innovation there?
- Current experiments only compare standard PT with TAPT. It would be interesting to see how other deep generative models perform for the same benchmark. I still believe established flow-based (or autoregressive networks) based approaches would lead to substantially similar (if not better) results.
- An analysis of computational time for PT and TAPT (including generating training data and training of the model) would very important to complement the analysis and assess the advantages of the model compared to standard MCMC approaches.

**Details Of Ethics Concerns:**

Not applicable.

---

> ### Author Response · Authors · 2025-12-03
> **Response to questions**
>
> > **#1.** Related to the weaknesses pointed above, I'd kindly ask the authors to elaborate more about their framework and its unique property which makes it different, and potentially more performant, compared to the approaches listed above.
>
>
>
> The cited works are all in the regime of continuous state spaces (molecular configurations, lattice field theories) and are primarily concerned with unbiased sampling of a target distribution. In contrast, TAPT is designed for discrete Ising systems and explicitly targets combinatorial optimization (3D spin glass, integer factorization), where finding ground states is NP-hard. Our main use of Parallel Tempering is as a rugged-landscape optimizer, not as a high-precision equilibrium sampler.
>
> More concretely, approaches such as temperature-steerable flows, Boltzmann Generators, and “Skipping the Replica Exchange Ladder with Normalizing Flows” learn continuous transports to improve sampling and, in the last case, modify the replica-swap step in PT. TAPT keeps the standard PT swap ladder intact and instead injects learned state proposals directly into each replica, using PT as a verifier to drive a ground-state search in discrete spaces. To the best of our knowledge, there is no prior work that (i) uses a $\beta$-conditioned neural equilibrium sampler as a global proposal mechanism inside PT and (ii) demonstrates consistent gains on genuinely hard discrete optimization benchmarks such as 3D spin glasses and Ising-encoded factorization. This combination of discrete setting, optimization focus, and generator–verifier coupling is the central novelty of our framework.
>
>
>
>
> > **#2.** The conditioning of the transformer on the inverse temperature seems to suggest that it is only able to interpolate between different values of $\beta$. Nevertheless, prior works seems to report that extrapolation is also possible. I was therefore wondering if this is an inherent limitation of transformers or if the authors have not tried extrapolation at all.
>
>
>
> This is an interesting idea. Sampling at low temperatures is challenging because obtaining equilibrium samples requires very long mixing times. In fact, at high $\beta$ values, equilibrium sampling effectively reduces to finding the ground state, which is known to be NP-hard for combinatorial optimization problems. While extrapolation to higher $\beta$ is possible in the transformer, the quality of the generated samples inevitably degrades as $\beta$ moves further outside the training range. An example of this behavior is shown in Figure 7, where, at lower temperatures (far from the training range), the transformer proposals are never accepted.
>
>
>
>
> > **#3.** Why do the authors picked an autoregressive transformer as a generative model instead of a Diffusion model, autoregressive neural network (e.g., PixelCNN) or a normalising flow?
> > This approach requires data for training while approaches like Boltzmann generators are data free. What is the advantage then?
>
>
> Our main contribution is the optimization framework: a generator–verifier scheme where a learned equilibrium sampler is plugged into PT to accelerate ground-state search in hard discrete Ising problems. The specific choice of generator (here, an autoregressive Transformer) is one concrete instantiation, not the core novelty.
>
> TAPT itself is generator-agnostic. Any model that can propose high-quality configurations at a given $\beta$ — diffusion, PixelCNN-style autoregressive networks, normalizing flows, or “data-free” Boltzmann generators — could in principle be used in place of the Transformer without changing the TAPT logic. Our results show that given a reasonably accurate generator, using it as a global proposal mechanism inside PT yields clear optimization gains; systematically swapping in alternative generators is natural follow-up work.
>
>
>
>
> > **#4.** Section 4 mostly focus on the metropolis step within the TAPT algorithm which ensure to retain detailed balance and other properties typical of MCMC methods. The analysis therein seems very similar to the idea proposed by Nicoli et al. (2020). Am I seeing it right or does your method bring some innovation there?
>
>
>
> In Section 4, we use the standard Metropolis acceptance rule for transformer proposals (Eq. 2) without the full Metropolis–Hastings correction, because our experiments focus on optimization rather than unbiased estimation of a distribution. However, since the transformer is autoregressive and provides exact proposal probabilities, TAPT can be turned into an unbiased sampler by adding the Hastings correction whenever exact sampling is required. By contrast, the work of Nicoli et al. (2020) relies on the Hastings correction, as their primary goal is sampling rather than optimization.

---

> > ### Author Response · Authors · 2025-12-03
> >
> > > **#5.** Current experiments only compare standard PT with TAPT. It would be interesting to see how other deep generative models perform for the same benchmark. I still believe established flow-based (or autoregressive networks) based approaches would lead to substantially similar (if not better) results.
> >
> > We agree that it would be interesting to compare different deep generators (e.g., flows or other autoregressive models) on the same benchmarks. In this work our focus is not on the specific architecture of the generator, but on the effect of adding learned global proposals to PT for optimization. IsingFormer is one concrete choice; in principle TAPT could use any suitable generator, and studying this dependence is an interesting direction for future work.
> >
> > > **#6.** An analysis of computational time for PT and TAPT (including generating training data and training of the model) would very important to complement the analysis and assess the advantages of the model compared to standard MCMC approaches.
> >
> >
> > We agree that a more explicit discussion of computational cost is useful. Appendix F already reports the wall–clock training times for all models (on a single RTX 6000 Ada GPU): roughly 3 days for the $50 \times 50$ 2D Ising model, about 1 day for the $L=10$ 3D spin glass, and between 2.5 and 10 hours for the factorization models, depending on bit–width. The cost of generating the high–temperature MCMC training data is of the same order as a small number of PT runs, since mixing is still relatively fast in that regime.
> >
> > TAPT is designed for the setting where a generator is reused many times: multiple independent optimization runs, or many instances from the same family (as in our factorization experiments). In this regime, the offline training and data–generation costs are amortized, and the relevant comparison is the per–run cost at evaluation time, where TAPT reaches lower energies than PT under the same sweep budget. For a single, one–off optimization on a small instance, plain PT can indeed be more appropriate.

---

### Official Review · Reviewer_cYx3 · 2025-10-23

**Soundness:** 2
**Presentation:** 2
**Contribution:** 1
**Rating:** 2
**Confidence:** 5

**Summary:**

This work introduces Transformer-Augmented Parallel Tempering (TAPT) which uses a decoder-only Transformer (called IsingFormer) to autoregressively generate proposals for Parallel Tempering that are conditioned on the inverse temperature. The problem settings investigated are sampling and optimization problems on Ising spin systems. The methods relies on a training phase based on samples at different temperatures. Training samples are, however, limited to the high-temperature regime. It is demonstrated that TAPT can interpolate within the range of training temperatures. Once trained, TAPT accelerates sampling and optimization substantially and demonstrates generalization capabilities on a factorization problem.

**Strengths:**

- The graphical presentation of results is well done.

- The integer factorization problem is interesting and appears novel in the machine learning community.

**Weaknesses:**

- Reliance on MCMC samples. Comparison of this method to MCMC methods like in Fig. 2 b and c are not insightfuls since they compare a method that was trained on ground truth samples (TAPT) to a method that is not trained on any samples and which has to generate such samples only based on the energy function. Concretely, in Fig 2c  one step of TAPT is compared to one step Gibbs sampling. Such a comparison is problematic since it neglects the costs for generating the MCMC samples on which TAPT was trained and the training costs. This reliance on training data limits TAPT to high temperatures and at low temperatures where sampling becomes increasingly hard the method actually switches to pure PT (see l293 ff). In l364 the authors acknowledge this shortcoming of not accounting for the TAPT training time when comparing optimization performances; they do, however, not mention the costs of generating the training data. They then claim that this is the case for "many neural optimizers" which is arguably wrong. A substantial amount of the work in neural combinatorial optimization is (1) generalizing over problems and (2) not using training data (see for instance, the two cited works by Sanokowski et al. and the references therein). Accordingly, the authors fail to correctly represent this aspect in section 2.2 where they write "These methods often directly optimize to find low-energy states and retrain models for different instances of a problem.". This is an imprecise or even miss-leading statement since actually, most works in the ML community do generalize over problems, e.g. the cited works by Sanokowski et al. and references therein.
The authors should factor in all costs of TAPT and do fair comparisons that investigate with which number of drawn samples it is more efficient to use TAPT than regular PT.

- In 2.1 it is claimed that: "Importantly, flows, diffusion models, RBMs, or alternative autoregressive samplers complement our approach: gains in generator fidelity will translate directly into higher acceptance and more effective nonlocal moves.". Why is this claim not underpinned by experiments?

- Unfortunately, the related work section misses any discussion of other works that augment MCMC like PT or SMC with neural samplers. Recent examples in the continuous domain are Chen, 2024 or Zhang 2025, the later of which is conceptually extremely close since it also augments PT.

- The results in Fig. 2 a indeed represent generalization capabilities, but it is entirely unclear how to judge these results: the method is not compared to any other machine learning method. It is not clear which degree of generalization is sufficient for downstream applications, i.e. these results lack meaningful context.

- Unbiasedness: the authors state in section 4 about TAPT that: "This rule is theoretically valid if the transformer samples from the Boltzmann distribution at β_r". If the prerequisite is that TAPT can yield unbiased samples at each temperature why is PT then needed at all? The authors state that: "Because the IsingFormer can compute the precise probability Pmodel(m) for any given state m due to its autoregressive nature, it enables the use of the full Metropolis-Hastings (MH) correction". This is given as a reason to use the autoregressive approach which is, of course, very expensive. But actually, the MH correction is not even applied since it is not needed for optimization. So the whole reasoning appears inconclusive: the method is not really a MCMC method with according asymptotic sampling guarantees since it does not mitigate biases via MH correction and hence it is not clear why the expensive autoregressive approach is used.
The impact of MH correction should be investigated in the optimization and the sampling scenario.

- The methods should be compared to other neural optimization methods on common sampling and optimization benchmarks, like the ones used in the cited papers by Sanokowski.

Chen, J., Richter, L., Berner, J., Blessing, D., Neumann, G., & Anandkumar, A. (2024). Sequential controlled langevin diffusions. arXiv preprint arXiv:2412.07081.

Zhang, L., Potaptchik, P., He, J., Du, Y., Doucet, A., Vargas, F., ... & Syed, S. (2025). Accelerated Parallel Tempering via Neural Transports. arXiv preprint arXiv:2502.10328.

**Questions:**

- Autoregressive spin-wise sample generation: does TAPT use one forward-pass for each individual proposed spin?

- In l365 it is stated that: "As expected, acceptance diminishes at the cold end beyond the training β range...". Is this shown anywhere?

- Was the generalization of TAPT over $\beta$ also investigated for the 3D Ising problem?

---

> ### Author Response · Authors · 2025-12-03
> **Response to weaknesses**
>
> > 1- Reliance on MCMC samples. Comparison of this method to MCMC methods like in Fig. 2 b and c are not insightfuls since they compare a method that was trained on ground truth samples (TAPT) to a method that is not trained on any samples and which has to generate such samples only based on the energy function. Concretely, in Fig 2c one step of TAPT is compared to one step Gibbs sampling. Such a comparison is problematic since it neglects the costs for generating the MCMC samples on which TAPT was trained and the training costs. This reliance on training data limits TAPT to high temperatures and at low temperatures where sampling becomes increasingly hard the method actually switches to pure PT (see l293 ff). In l364 the authors acknowledge this shortcoming of not accounting for the TAPT training time when comparing optimization performances; they do, however, not mention the costs of generating the training data. They then claim that this is the case for "many neural optimizers" which is arguably wrong. A substantial amount of the work in neural combinatorial optimization is (1) generalizing over problems and (2) not using training data (see for instance, the two works by Sanokowski et al. and the references therein). Accordingly, the authors fail to correctly represent this aspect in section 2.2 where they write "These methods often directly optimize to find low-energy states and retrain models for different instances of a problem.". This is an imprecise or even miss-leading statement since actually, most works in the ML community do generalize over problems, e.g. the works by Sanokowski et al. and references therein.
>
> **W1:**
> MCMC is guaranteed to sample from the Boltzmann distribution, but the bottleneck is mixing time. Figure 2 is explicitly designed to evaluate *fidelity*: we compare the samples produced by the transformer to long-run MCMC, which we treat as ground truth. We do not present TAPT as a competitive sampler for the 2D Ising model. Instead, 2D Ising serves as a controlled testbed to check whether the transformer can capture thermodynamic limits and global structure.
>
> Regarding cost: training and data generation are indeed nontrivial, but these costs are amortized when the model is reused across many queries and instances. We show this explicitly in the factorization task, where a single trained model generalizes to unseen instances from the same family and improves optimization performance.
>
> TAPT does not "switch back" to plain PT at low temperatures. While the transformer does not make proposals for the very coldest replicas (where equilibrium data are intractable to obtain), high quality proposals at intermediate temperatures do propagate down the temperature ladder and still accelerate ground state search at the lowest temperatures.
>
> Finally, our intent in Sec. 2.2 was not to suggest that *no* neural combinatorial optimization methods generalize across instances. There is an active line of work that aims exactly at this, including the discrete diffusion models of Sanokowski et al. However, robust instance-level generalization across problem sizes and disorder realizations remains a central challenge for neural samplers, and this is one reason they are not yet used as standard heuristics in the way that classical MCMC or PT are.
> Our goal in Sec. 2.2 was to highlight a different axis: TAPT is a proposal-augmentation layer that can sit on top of any generator, whether or not that generator generalizes across instances. In principle, discrete diffusion generators such as those of Sanokowski et al. could be used inside TAPT in place of the transformer. Our results additionally show distributional agreement with long-run MCMC at fixed $\beta$, not only low-energy solutions, which is a stronger notion than just finding good minima on held-out instances.
>
> > 2-In 2.1 it is claimed that: "Importantly, flows, diffusion models, RBMs, or alternative autoregressive samplers complement our approach: gains in generator fidelity will translate directly into higher acceptance and more effective nonlocal moves.". Why is this claim not underpinned by experiments?
>
> **W2:**
> While it would be interesting to evaluate TAPT’s performance under different proposal generators, it is virtually impossible to explore all such choices in a single coherent work. The point we wanted to make in Sec. 2.1 is conceptual: in TAPT, any generator that proposes configurations closer to the Boltzmann distribution at a given $\beta_r$ will, by construction of the Metropolis–Hastings rule, lead to higher acceptance rates and more effective nonlocal moves. In this paper we instantiate the generator with a Transformer because it is a natural fit for discrete Ising systems and provides tractable log probabilities. Systematically swapping in flows, diffusion models, or RBMs is an orthogonal, and quite large, experimental axis; we view this as follow-up work rather than something we can reasonably include in this first paper.

---

> > ### Author Response · Authors · 2025-12-03
> >
> > > 3- Unfortunately, the related work section misses any discussion of other works that augment MCMC like PT or SMC with neural samplers. Recent examples in the continuous domain are Chen, 2024 or Zhang 2025, the later of which is conceptually extremely close since it also augments PT.
> >
> >
> > **W3:**
> >
> > The suggested works by Chen (2024) and Zhang (2025) operate in continuous state spaces and address continuous-time sampling, whereas our paper focuses on discrete Ising spin systems and combinatorial optimization; these are fundamentally different problem classes, so we did not treat those methods as primary baselines or closely related work.
> >
> > > 4- The results in Fig. 2 a indeed represent generalization capabilities, but it is entirely unclear how to judge these results: the method is not compared to any other machine learning method. It is not clear which degree of generalization is sufficient for downstream applications, i.e. these results lack meaningful context.
> >
> >
> > **W4:**
> >
> > Figure 2(a) is intended as a sanity check that the transformer generalizes across inverse temperatures within its training range and matches thermodynamic curves, not as a benchmark of "best possible" ML model on 2D Ising.
> >
> > > 5-Unbiasedness: the authors state in section 4 about TAPT that: "This rule is theoretically valid if the transformer samples from the Boltzmann distribution at $\beta_r$". If the prerequisite is that TAPT can yield unbiased samples at each temperature why is PT then needed at all? The authors state that: "Because the IsingFormer can compute the precise probability Pmodel(m) for any given state m due to its autoregressive nature, it enables the use of the full Metropolis-Hastings (MH) correction". This is given as a reason to use the autoregressive approach which is, of course, very expensive. But actually, the MH correction is not even applied since it is not needed for optimization. So the whole reasoning appears inconclusive: the method is not really a MCMC method with according asymptotic sampling guarantees since it does not mitigate biases via MH correction and hence it is not clear why the expensive autoregressive approach is used. The impact of MH correction should be investigated in the optimization and the sampling scenario.
> >
> > **W5:**
> >
> > For the *optimization* experiments that are the main focus of this paper, we do not need unbiased sampling from the Boltzmann distribution. We are interested in finding low energy states, not in computing exact expectations, so we use TAPT as a proposal mechanism inside PT without MH on top of PT itself. In this regime the principal constraints are wall clock time and solution quality, and the autoregressive cost is comparable to a PT sweep for the system sizes considered.
> >
> > We still choose an autoregressive model for two reasons: (i) it is a natural choice for discrete Ising systems and gives normalized probabilities for all states, which enables an exact MH scheme when unbiased sampling is required, and (ii) it keeps the interface simple for future extensions to exact or approximate sampling.
> >
> >
> > > 6- The methods should be compared to other neural optimization methods on common sampling and optimization benchmarks, like the ones used in the cited papers by Sanokowski.
> >
> > **W6:**
> >
> > We do not agree that this work should be required to compare against all existing ML-based optimizers. The space of neural combinatorial optimization methods is large and evolving, and an exhaustive comparison is not realistic in a single paper. Our goal here is narrower and more concrete: to show that augmenting a very strong physics baseline (adaptive PT) with learned global proposals can yield clear gains on hard spin-glass and factorization problems.
> >
> > We view a systematic benchmark against multiple neural optimizers as a separate, follow-up study. TAPT is explicitly designed as a proposal-augmentation layer, so in principle those methods could be used as generators inside TAPT rather than as competitors in a one-off leaderboard comparison.

---

> ### Author Response · Authors · 2025-12-03
> **Response to questions**
>
> > **#1.** Autoregressive spin-wise sample generation: does TAPT use one forward-pass for each individual proposed spin?
>
>
>
> Yes, generating a proposal uses a single forward pass per spin, which we use to generate the full spin configuration.
>
>
>
> > **#2.** In l365 it is stated that: "As expected, acceptance diminishes at the cold end beyond the training $\beta$ range...". Is this shown anywhere?
>
> Figure 7(c) in Appendix C shows that proposals for the coldest replica (indexed as replica 22) are never accepted. The training range for $\beta$ is reported in Appendix F ($\beta \in [0.125, 2.0]$). However, we acknowledge the reviewer’s point that we did not specify the $\beta$ vector used for TAPT in the 3D spin-glass experiment, which differs from the training values. In this experiment, TAPT used 22 replicas, with a temperature range of $\beta \in [0.5, 5.8]$.
>
> > **#3.** Was the generalization of TAPT over $\beta$ also investigated for the 3D Ising problem?
>
> We tested the transformer at multiple $\beta$ values that were not included in training. The results show a good match with MCMC at low $\beta$ as illustrated in Fig.2 in the supplementary. We note that such overlap was not observed at higher $\beta$ values

---

### Official Review · Reviewer_QAcf · 2025-10-29

**Soundness:** 2
**Presentation:** 1
**Contribution:** 2
**Rating:** 2
**Confidence:** 4

**Summary:**

A transformer is trained on equilibrium spin configurations. It is used to propose global moves that aim to improve Parallel Tempering (PT) and evaluated on a simple spin-glass model and an Ising-encoded factorization task.

**Strengths:**

The idea of using a transformer as a global proposal generator in PT is interesting and conceptually sound. It is relevant to test whether such models can learn meaningful structure, and applying it to both a simple spin glass and a more applied factorization problem is a reasonable choice.

**Weaknesses:**

The presentation is unclear. The description of the transformer and how it is integrated into PT is scattered, and architectural details appear only later in the paper (Appendix F), after results are already discussed in Section 3.

The experiments do not convincingly show that the transformer learns structural correlations beyond global thermodynamic properties.

The benefit of the transformer appears limited. As shown in Figure 7, adding context reduces performance, suggesting that the model generates only global thermodynamically consistent proposals without capturing the spatial correlations present in the training data.

Benchmarks are limited: few problem instances, no comparison to other global or learned-move algorithms, and no study of scaling with problem size.

The ablation studies are not fully convincing. The influence of architectural parameters (number of layers, heads, width) is not analyzed.

There appears to be no control replacing transformer proposals with random global perturbations. Without such tests, it is unclear whether improvements come from learned structure or simply from periodic global resets.

**Questions:**

1) The model is trained on equilibrium samples to learn the equilibrium distribution, but only for higher temperatures. The claim is that low-temperature data are not helpful. Could you show performance (e.g., residual energy after fixed steps) as a function of the lowest temperature where the transformer is applied? Does performance saturate at a certain temperature depth, and does that correspond to the spin-glass transition or deeper in the glassy regime?

2) How does the transformer architecture affect performance? Please report results versus number of parameters, hidden size, attention heads, and layers.

3) A useful ablation study would be to replace the transformer’s global move by a random global perturbation? This would test whether the transformer learns something meaningful or merely acts as a reset. To isolate the benefit of learned structure, include controls using random or β-conditioned global resets at identical frequency.

4) For the combinatorial optimization experiments, computing residual energy on planted instances (e.g., as in arXiv:1906.00275) would provide an unambiguous ground-state reference.

5) The appendix derivation of the 2D Ising free energy appears to reproduce classical results rather than new material. Please clarify whether this is included only for completeness or if any modification is introduced ?

---

> ### Author Response · Authors · 2025-12-03
> **Response to weaknesses**
>
> > 1- The presentation is unclear. The description of the transformer and how it is integrated into PT is scattered, and architectural details appear only later in the paper (Appendix F), after results are already discussed in Section 3.
>
> **1. Presentation.** We intentionally placed specific architectural details in **Appendix F** because TAPT is a **generator-agnostic framework**. The manuscript prioritizes the algorithmic methodology and physical validation (Section 3) over implementation specifics. Focusing on one particular Transformer architecture early in the text would distract from the core contribution: the integration of *any* learned generator into the Parallel Tempering verification loop. In the revision, we can clarify this design choice early on and add an explicit pointer to Appendix F for readers who want architectural details.
>
> > 2-The experiments do not convincingly show that the transformer learns structural correlations beyond global thermodynamic properties.
>
> **2. Experiments.** We are not sure what “learning structural correlations” is intended to mean here. The purpose of the transformer in our algorithm is to learn the global thermodynamic properties of glassy systems. Obtaining these exactly for generic models is #P-hard. Our core contribution is to show that a transformer that has learned these global thermodynamic properties can help the powerful PT algorithm.
>
> > 3- The benefit of the transformer appears limited. As shown in Figure 7, adding context reduces performance, suggesting that the model generates only global thermodynamically consistent proposals without capturing the spatial correlations present in the training data.
>
> **3. Benefit.** We agree that context not helping is an interesting feature. We did show that correct context completion by the transformer is possible, but optimization performance is only improved with totally independent samples. In other words, the best use of the model here is to generate independent, thermodynamically consistent proposals rather than conditional refinements. We still observe a clear net benefit with transformers; the reviewer seems to be critiquing the “mechanism” without acknowledging the empirical benefit.
>
> > 4- Benchmarks are limited: few problem instances, no comparison to other global or learned-move algorithms, and no study of scaling with problem size.
>
> **4. Benchmarks.** Adaptive Parallel Tempering is considered to be the gold standard when it comes to optimizing unstructured 3D spin glasses. It is virtually impossible in a single paper to compare a new optimization algorithm with all other possible “learned-move” algorithms. As a first step, we believe that our comparison to adaptive PT on a 3D spin glass and an Ising-encoded factorization task constitutes new knowledge. Extending this to further benchmark families and additional learned samplers is a natural direction for follow-up work.
>
>
> > 5-The ablation studies are not fully convincing. The influence of architectural parameters (number of layers, heads, width) is not analyzed.
>
> **5. Ablation studies.** The reviewer points out that we do not vary the internal architectural parameters of the transformer (number of layers, heads, width). This is by design. We deliberately keep the architecture small and fixed and focus our ablations on the aspects specific to TAPT: for example, the number of TAPT replicas and the amount of context provided to the model. Exploring wider or deeper architectures is orthogonal to the main message of this work and is a natural direction for future scaling studies; our goal here is to show that even a compact generator can already improve PT.
>
> > 6- There appears to be no control replacing transformer proposals with random global perturbations. Without such tests, it is unclear whether improvements come from learned structure or simply from periodic global resets.
>
> **6. Control: random global perturbations.** The manuscript already mentions that using a transformer trained for one disorder instance on a *different* instance completely fails: proposals are almost never accepted and TAPT worsens even standard PT. This indicates that the transformer proposals are highly instance-specific and not just random global perturbations. We also show in Fig. 1 (added to the supplementary) that simple global random resets hurt optimization performance rather than benefit it. In contrast, transformer proposals with high acceptance probability consistently improve over PT.

---

> ### Author Response · Authors · 2025-12-03
> **Response to questions**
>
> > **#1.** The model is trained on equilibrium samples to learn the equilibrium distribution, but only for higher temperatures. The claim is that low-temperature data are not helpful. Could you show performance (e.g., residual energy after fixed steps) as a function of the lowest temperature where the transformer is applied? Does performance saturate at a certain temperature depth, and does that correspond to the spin-glass transition or deeper in the glassy regime?
>
>
> We do not claim that low-temperature data are not helpful; if anything, they would be extremely informative. The issue is that obtaining *equilibrium* samples at very high $\beta$ (low temperature) is computationally intractable for the 3D instances we study, as we already state in the manuscript. In practice, when we try to apply the generator too deep in the glass phase, almost all proposed moves are rejected by the Metropolis rule, so these replicas effectively behave like plain PT. This is exactly what is visible in Fig. 7(c), where the acceptance rate drops to almost zero at the lowest temperatures beyond the transformer’s training range.
>
> Regarding the suggested experiment: varying the number of transformer-augmented replicas $N_T$ in Fig. 7 already probes how “deep” in temperature we can apply the transformer. As $N_T$ increases, the residual energy improves, but the gain quickly saturates once the augmented replicas reach the upper end of the training range in $\beta$. Beyond that, additional colder replicas see almost no accepted transformer moves, so performance no longer changes. In a revision we can make this dependence more explicit by plotting residual energy as a function of the minimum augmented temperature; this curve essentially tracks the acceptance profile and is controlled by where transformer proposals remain non-negligibly accepted, not by pushing arbitrarily deep into the glassy regime where equilibrium data cannot be obtained.
>
>
>
> > **#2.** How does the transformer architecture affect performance? Please report results versus number of parameters, hidden size, attention heads, and layers.
>
>
> We provide a small example on 20-bit factorization (310 spins) where we vary the transformer size and measure how often TAPT improves over PT. We change the model dimension, number of heads, and number of layers, while keeping the TAPT procedure fixed. As shown below, a larger transformer gives a clear gain on both training and test instances.
>
>
> | Instance Type        | $d_{model}$ | Number of Heads | Number of Layers | FFN Dimension      | TAPT on Training Instances | TAPT on Testing Instances |
> |----------------------|---------|-----------------|------------------|--------------------|----------------------------|---------------------------|
> | 20-bit Factorization | 64      | 2 | 2 | 2 $\times d_{model}$          | ~61%   | ~55%         |
> | 20-bit Factorization | 256     | 4 | 5 | 4 $\times d_{model}$          | ~94%  | ~90%         |
>
>
>
> > **#3.** A useful ablation study would be to replace the transformer’s global move by a random global perturbation? This would test whether the transformer learns something meaningful or merely acts as a reset. To isolate the benefit of learned structure, include controls using random or $\beta$-conditioned global resets at identical frequency.
>
>
>
> Please see our previous response that includes the global resetting experiment.
>
>
> > **#4.** For the combinatorial optimization experiments, computing residual energy on planted instances (e.g., as in arXiv:1906.00275) would provide an unambiguous ground-state reference.
>
>
>
> We agree. For the 3D spin glass we follow standard practice and use the same best-known ground-state reference for both PT and TAPT, so both methods are compared based on the same relative energy performance. Also, for factorization, the ground state is known exactly by construction (the true factors of the semiprime), so it is already unambiguous and is an example of a planted instance.
>
> > **#5.** The appendix derivation of the 2D Ising free energy appears to reproduce classical results rather than new material. Please clarify whether this is included only for completeness or if any modification is introduced ?
>
>
> The appendix is not trying to introduce a new solution to the 2D Ising model. The classical Onsager result is, of course, used. What is new in the appendix is the part we actually need for our experiment: the exact free energy of a finite $50\times 50$ system with a clamped top boundary. This is not directly available from the standard closed form.
>
> The appendix simply shows how we adapt the classical planar-Ising machinery to this clamped geometry so that the benchmark in Fig. 2(c) is fully specified. We do not present this as a major theoretical result. It is included because the clamped free-energy calculation is not something one can pull straight from a textbook.

---

### Official Review · Reviewer_tQRd · 2025-10-31

**Soundness:** 2
**Presentation:** 3
**Contribution:** 2
**Rating:** 2
**Confidence:** 4

**Summary:**

The paper is proposing  a new strategy to improve parallel tempering (PT) algorithm, based on using a transformer model
in order to propose global moves to high temperatures replicas of PT.
The method is demonstrated  on small size Ising 2D, a 3D spin glass model and an integer factorization problem.

**Strengths:**

- the sampler is able to generalize at intermediate unseen temperatures without retraining and shows good performances with respect to PT in optimization tasks.

**Weaknesses:**

- Novelty: The paper claims the application of a new Generator-Verifier framework to sampling and combinatorial optimization problems. However it looks quite similar to already proposed architectures[1,2] (except for the Transformer architecture part), yet not applied to optimization tasks, which also involve auto-regressive models to perform  global moves. The novelty of the present method with respect to those resides essentially in the use of a transformer for the autoregressive model.
 - clarity: some details (is there any available repository for the implementation?) are missing to understand what the transformer is actually doing, some specification are given in appendix but not much explained,  in particular how the spins are tokenized is not clear. Also the training method of the transformer model is not clearly stated, so it is not clear how to reproduce the results.
- Evaluation: the sampling method is only compared against vanilla PT. The impact of the architecture of the auto-regressive model could be assessed by  comparing to[1,2]. Also in terms of computational efficiency, the methods is likely very heavy, and no comparison is given.

References:
[1] Causer L, Rotskoff GM, Garrahan JP. Discrete generative diffusion models without stochastic differential equations: A tensor network approach. Physical Review E. 2025 Feb;111(2):025302.
[2] Del Bono LM, Ricci-Tersenghi F, Zamponi F. On the performance of machine-learning-assisted Monte Carlo in sampling from simple statistical physics models. arXiv e-prints. 2025 May:arXiv-2505.

**Questions:**

More specifically I have the following questions:

 - Is there some specificity to the framework proposed that makes it different from proposing global moves to intermediate temperatures of a PT sampling scheme ?
 - How is training performed: is there a tokenization taking place at some point ? Is a specific ordering of the spins used ?
 - In the plain text, it is mentioned that Metropolis-Hasting corrections are performed (L 278-285). However, lines 10-11 of Algorithm 1 clearly perform this step,  and L 050 of the main text include the MH step as part of the verifier. Is the MH step actually performed or not? If not, what are the guarantees of actually sampling the equilibrium  distribution of the model ? For instance for the reports of the free energy per spin in Table 1, since the model is Ising 2d, it is made of two potential wells with the exact same depth,  so the goodness of the free energy per spin only ensures that a local convergence at least one of the well is obtained but it does not tell whether perfect balance between the two wells was obtained. Stated differently, mode dropping could occur without notice in this case.
 - how does the algorithm scale with size? for instance the size 50x50 is very small compared to what can be done with a dedicated algorithm like Wolff algorithm which is also able to  propose global moves by flipping clusters of spins. Is it possible to obtain reasonable scaling function and critical exponents by varying the size of the system on the 2D Ising with the proposed algorithm  for instance?
 - how good are the performances in the optimization, could we have some elements of comparison with dedicated solver in terms of precision and computational time?

I would suggest to have more extensive tests (rather than on the 2D Ising model, which is not so informative) to validate that the generator actually learn equilibrium physics in particular on multi-modal dataset with not necessarily well balanced modes.

---

> ### Author Response · Authors · 2025-12-03
> **Response to weaknesses**
>
> We thank the reviewer for their comments. We believe there is a fundamental distinction regarding the scope of our work compared to the cited references. While References [1, 2] utilize generative models for **sampling** equilibrium distributions, our work focuses on **combinatorial optimization** for NP-hard problems (3D Spin Glasses and Integer Factorization). Crucially, our algorithm (TAPT) demonstrates **generalization** across problem instances, a capability not typically addressed in standard sampling literature.
>
> We address the specific points below.
>
> > 1- Novelty: The paper claims the application of a new Generator-Verifier framework to sampling and combinatorial optimization problems. However it looks quite similar to already proposed architectures[1,2] (except for the Transformer architecture part), yet not applied to optimization tasks, which also involve auto-regressive models to perform global moves. The novelty of the present method with respect to those resides essentially in the use of a transformer for the autoregressive model.
>
> **1. Novelty:** The reviewer suggests our framework is similar to [1, 2] but ``not applied to optimization tasks.'' This distinction is central to our contribution:
>
> - **Sampling vs. Combinatorial Optimization:** The cited references use generative models to sample from Boltzmann distributions. TAPT explicitly bridges learned sampling and hard optimization. We use the generator to propose uncorrelated global moves that allow the verifier (PT) to escape deep local minima in rugged landscapes, rather than strictly aiming for unbiased sampling statistics.
>
> - **Generalization:** A key novelty of TAPT is training on a subset of problem instances (e.g., semiprimes) and generalizing to **unseen instances** without retraining. This is critical for neural combinatorial optimization. It is also a currently unsolved problem: neural optimizers, by and large, do not play any role in real-world optimization.
>
> - **Generator-Agnostic:** TAPT is a framework that couples a global proposer with a physics-based verifier. If the architectures in [1, 2] offer superior sampling for specific domains, they can be integrated into TAPT.
>
> > 2-clarity: some details (is there any available repository for the implementation?) are missing to understand what the transformer is actually doing, some specification are given in appendix but not much explained, in particular how the spins are tokenized is not clear. Also the training method of the transformer model is not clearly stated, so it is not clear how to reproduce the results.
>
> **2. Clarity**: The details requested by the reviewer are explicitly provided in the text:
>
> - **Tokenization:** As shown in **Figure 1**, the lattice is unwrapped into a 1D binary sequence. The Transformer models this autoregressively via next-token prediction using a standard vocabulary $v \in$ {0,1}.
>
> - **Training:** Detailed in **Appendix F** (lines 920-971). We train using a binary cross-entropy loss on equilibrium samples generated by long-run MCMC.
>
> - **Reproducibility:** We disclosed the architecture (decoder-only, $d_{model}=64$, 2 layers) and hardware (RTX 6000 Ada). We plan to release our code upon publication.
>
> > 3- Evaluation: the sampling method is only compared against vanilla PT. The impact of the architecture of the auto-regressive model could be assessed by comparing to[1,2]. Also in terms of computational efficiency, the methods is likely very heavy, and no comparison is given.
>
> **3. Evaluation:**  Comparing TAPT (an optimizer for NP-hard problems) against pure samplers does not capture the metric of interest: finding ground states in rugged landscapes.
> The reviewer notes the method is computationally heavy. We address this via **amortization** in **Section 5.1** and **Appendix F**. While training has an upfront cost, we train *once* (e.g., $\sim$2.5 hours for 8-bit factorization) and solve many *unseen* problems rapidly. A single inferred proposal replaces thousands of local MCMC updates, which is a key contribution of our work.

---

> ### Author Response · Authors · 2025-12-03
> **Response to questions**
>
> > **More specifically I have the following questions:**
> > **#1.** Is there some specificity to the framework proposed that makes it different from proposing global moves to intermediate temperatures of a PT sampling scheme ?
>
> Precisely: the integration of learned, autoregressive global moves into the PT framework *is* the core novelty. While the concept of global moves (e.g., cluster updates) exists, they famously fail in frustrated systems like spin glasses. The “specificity” of our framework is the use of a Transformer to generate **statistically independent, uncorrelated samples**. This allows replicas to effectively “teleport” across the energy landscape rather than diffusing through it. To our knowledge, this specific mechanism of using a learned autoregressive model to drive global jumps within a PT verifier for hard optimization is a new idea.
>
>
>
> > **#2.** How is training performed: is there a tokenization taking place at some point ? Is a specific ordering of the spins used ?
>
> We use a simple binary vocabulary with two tokens {0,1}, where each token corresponds to a single spin. We train the transformer autoregressively with a standard binary cross-entropy loss on equilibrium configurations at few high-temperatures (conditioning on $\beta$). A fixed spin ordering is used for each problem: for 2D Ising we use a zigzag order of the lattice; for factorization the spins are ordered as {$\text{product}$}\{$\text{carry bits}$}{$\text{digital multiplier}$}{$\text{factor B}$}{$\text{factor A}$}.
>
>
> > **#3.** In the plain text, it is mentioned that Metropolis-Hasting corrections are performed (L 278-285). However, lines 10-11 of Algorithm 1 clearly perform this step, and L 050 of the main text include the MH step as part of the verifier. Is the MH step actually performed or not? If not, what are the guarantees of actually sampling the equilibrium distribution of the model ? For instance for the reports of the free energy per spin in Table 1, since the model is Ising 2d, it is made of two potential wells with the exact same depth, so the goodness of the free energy per spin only ensures that a local convergence at least one of the well is obtained but it does not tell whether perfect balance between the two wells was obtained. Stated differently, mode dropping could occur without notice in this case.
>
>
> To clarify: we use the standard **Metropolis** acceptance rule (Eq. 2), based solely on energy differences. We do *not* apply the full **Hastings** correction (which would require the ratio of autoregressive probabilities). Algorithm 1 (Line 10) is consistent with the text: it calculates acceptance based on $\Delta E$, not proposal densities.
>
> While strict detailed balance is required for unbiased sampling, our primary goal is **optimization**. The Metropolis rule sufficiently filters high-energy proposals, while the subsequent local MCMC updates refine the accepted states.
>
> Regarding mode dropping: The Transformer successfully learns the global symmetry of the 2D Ising model. As shown below, the empirical probabilities of the degenerate ground states (all-zero and all-ones) from $10^5$ independent transformer samples are nearly identical, confirming that the model does not collapse into a single well:
>
> \begin{array}{c|cc}
> \beta & P(0\ldots 0) & P(1\ldots 1) \\\\
> \hline
> 0.875 & 0.03 & 0.03 \\\\
> 1.000 & 0.20 & 0.19 \\\\
> 1.125 & 0.34 & 0.33 \\\\
> \end{array}
>
>
>
> > **#4.** how does the algorithm scale with size? for instance the size 50x50 is very small compared to what can be done with a dedicated algorithm like Wolff algorithm which is also able to propose global moves by flipping clusters of spins. Is it possible to obtain reasonable scaling function and critical exponents by varying the size of the system on the 2D Ising with the proposed algorithm for instance?
>
> We agree that for unfrustrated ferromagnetic models, specialized cluster algorithms (e.g., Wolff) are highly efficient. However, these methods are totally ineffective for our combinatorial optimization  applications (**3D Spin Glasses**,  **Integer Factorization** and beyond) where frustration prevents the formation of flipable clusters.
>
> We utilize the 50$\times$50 2D Ising model strictly as a **validation benchmark**. Because the exact solution is known, it allows us to quantitatively verify that the generator captures global structures (e.g., magnetization curves and energy barriers) before applying it to intractable frustrated problems.  Our broader goal is to demonstrate that learned proposals can accelerate optimization in rugged landscapes where cluster moves are not an option.

---

> ### Author Response · Authors · 2025-12-03
>
> > **#5.** How good are the performances in the optimization, could we have some elements of comparison with dedicated solver in terms of precision and computational time?
>
>
> For 3D Spin Glasses, Parallel Tempering *is* the gold standard and the widely accepted state-of-the-art method for ground state search. There are no specialized exact solvers for 3D spin glasses that scale efficiently. Therefore, demonstrating acceleration over PT (as shown in Figure 3) constitutes a direct comparison against the primary benchmark in the field.
>
> Regarding Integer Factorization, we explicitly treat it as a canonical hard Ising benchmark rather than a competition with number-theoretic algorithms (like the Number Field Sieve). Our goal is to demonstrate that TAPT accelerates generic Ising optimization where standard PT struggles.
>
>
>
> > **#6.** I would suggest to have more extensive tests (rather than on the 2D Ising model, which is not so informative) to validate that the generator actually learn equilibrium physics in particular on multi-modal dataset with not necessarily well balanced modes.
>
>
>
> For NP-hard problems like 3D Spin Glasses and Integer Factorization, strictly validating equilibrium learning (e.g., via partition functions) is **computationally intractable**. If we could compute exact thermodynamics to verify the generator, we would have already solved the problem. The 2D Ising model was included precisely because it is the only rigorous testbed where such validation is possible.
>
> For hard instances, the generator's goal is not perfect unbiased sampling, but **optimization advantage**: learning enough global structure to propose moves that escape deep local minima. Our experiments on 3D Spin Glasses (Fig. 3) and Factorization (Fig. 4), which are inherently multi-modal and unbalanced, demonstrate exactly this. The fact that TAPT accelerates ground-state search and generalizes to unseen instances confirms the model captures the relevant physics required to assist the optimizer.

---

### Meta-Review · Area_Chair_kNKK · 2025-12-26

**Summary:**

The paper is proposing a new strategy to improve parallel tempering (PT) algorithm.  A transformer model is trained on the equilibirum distributions of high temperature chains of PT and used to generate uncorrelated samples, leadig to faster mixing. The method is experimentally tested on small size Ising model, a 3D spin glass model, and an integer factorization problem.

**Reviewer Concerns:**

The reviewers consistently raised concerns about the paper’s novelty, clarity, and experimental scope. Some of them noted strong similarities to prior generator-assisted or learned-proposal methods. Additional critiques focused on unclear presentation and reproducibility, limited experimental comparisons. Moreover, reviewers noted a  lack of runtime, scalability, and cost analyses, especially given that equilibrium samples have to be  optained first for the training of the  transformer.
In the rebuttal, the authors provided clear conceptual clarifications, emphasising that their controniton is an optimization-focused, generator-agnostic framework and not an unbiased sampling method and that they focus on sampling from discrete not continous state spaces. Furthermore, they explained design choices and training procedures. While these replies partially address concerns about clarity and positioning, several reviewers’ requests—most notably stronger evidence of novelty, broader baselines, and more thorough efficiency evaluations—remain only partially resolved. Overall, the rebuttal improves understanding of the work but does not fully dispel doubts about its incremental contribution and practical advantages which could be resolved in a resbumission of the work.

**Reviewer Scores:**

Three raviewers gave 2 as a score and only one gave a 4. The authors replied to all questions in the rebuttal, but I dont expect that the reply would change the general option of the reviewers that much that they they would vote for acceptance.

---

### Decision · Program_Chairs · 2026-01-26

Reject